



# Contributions of biomass-burning, urban, and biogenic emissions to the concentrations and light-absorbing properties of particulate matter in central Amazonia during the dry season

Suzane S. de Sá (1), Luciana V. Rizzo (2), Brett B. Palm[a] (3), Pedro Campuzano-Jost (3),

Douglas A. Day (3), Lindsay D. Yee (4), Rebecca Wernis (5), Gabriel Isaacman-VanWertz[b] (4),

Joel Brito[c] (6), Samara Carbone[d] (6), Yingjun J. Liu[e] (1), Arthur Sedlacek (7), Stephen

Springston (7), Allen H. Goldstein (4), Henrique M. J. Barbosa (6), M. Lizabeth Alexander (8),

Paulo Artaxo (6), Jose L. Jimenez (3), Scot T. Martin[*] (1,9)

(1) John A. Paulson School of Engineering and Applied Sciences, Harvard University, Cambridge, Massachusetts, USA
(2) Department of Environmental Sciences, Universidade Federal de São Paulo, Diadema, São Paulo, Brazil
(3) Department of Chemistry and Cooperative Institute for Research in Environmental Sciences, University of Colorado, Boulder, Colorado, USA
(4) Department of Environmental Science, Policy, and Management, University of California, Berkeley, Berkeley, California, USA
(5) Department of Civil and Environmental Engineering, University of California, Berkeley, Berkeley, California, USA
(6) Institute of Physics, University of São Paulo, São Paulo, Brazil
(7) Brookhaven National Laboratory, Upton, New York, USA
(8) Environmental Molecular Sciences Laboratory, Pacific Northwest National Laboratory, Richland, Washington, USA
(9) Department of Earth and Planetary Sciences, Harvard University, Cambridge, Massachusetts, USA
[a] Now at Department of Atmospheric Sciences, University of Washington, Seattle, USA
[b] Now at Department of Civil and Environmental Engineering, Virginia Tech, Blacksburg, Virginia, USA
[c] Now at IMT Lille Douai, Université Lille, SAGE, Lille, France
[d] Now at Agrarian Sciences Institute, Federal University of Uberlândia, Minas Gerais, Brazil
[e] Now at College of Environmental Science and Engineering, Peking University, Beijing, China

[*]To Whom Correspondence Should be Addressed

*E-mail: scot_martin@harvard.edu*

*https://martin.seas.harvard.edu/*



**Abstract**

2       Urbanization and deforestation have important impacts on atmospheric particulate matter

(PM) over Amazonia. This study presents observations and analysis of submicron $PM_1$
concentration, composition, and optical properties in central Amazonia during the dry season.
The focus is on delineating the anthropogenic impact on the observed quantities, especially as
related to the organic $PM_1$. The primary study site was located 70 km to the west of Manaus, a
city of over two million people in Brazil. As part of the GoAmazon2014/5 experiment, datasets
from a large suite of instrumentation were employed. A high-resolution time-of-flight aerosol
mass spectrometer (AMS) provided data on $PM_1$ composition, and aethalometer measurements
were used to derive the absorption coefficient $b_{abs,BrC}$ of brown carbon (BrC) at 370 nm. The
relationships of $b_{abs,BrC}$ with AMS-measured quantities showed that the absorption was
associated with less-oxidized, nitrogen-containing organic compounds. Atmospheric processing
appeared to bleach the BrC components. The organic $PM_1$ was separated into different classes by
positive-matrix factorization (PMF). Estimates of the effective mass absorption efficiency
associated with each PMF factor were obtained. Biomass burning and urban emissions appeared
to contribute at least 80% of $b_{abs,BrC}$ while accounting for 30 to 40 % of the organic $PM_1$ mass
concentration. In addition, a comparison of organic $PM_1$ composition between wet and dry
seasons revealed that only a fraction of the nine-fold increase in mass concentration between the
seasons was due to biomass burning. An eight-fold increase in biogenic secondary organic $PM_1$
was observed. A combination of decreased wet deposition and increased emissions and oxidant
concentrations, as well as a positive feedback on larger mass concentrations are thought to play a
role in the observed increases. Fuzzy c-means clustering identified three clusters to represent
different pollution influences during the dry season, including "baseline" (dry season
background, which includes biomass burning), "event" (increased influence of biomass burning
and long-range transport of African volcanic emissions), and "urban" (Manaus influence on top



of the background). The baseline cluster was associated with a mean mass concentration of $9 \pm 3$
$\mu g\ m^{-3}$. This concentration increased on average by 3 $\mu g\ m^{-3}$ for both the urban and the event
clusters. The event cluster was characterized by remarkably high sulfate concentrations.
Differences in the organic $PM_1$ composition for the urban cluster compared to the other two
clusters suggested a shift in oxidation pathways as well as an accelerated oxidation cycle due to
urban emissions, in agreement with findings for the wet season.



## 1. Introduction

The Amazon basin has undergone significant urbanization and deforestation in the past
decades (Davidson et al., 2012; Martin et al., 2017; van Marle et al., 2017). An understanding of
how the composition of atmospheric particulate matter (PM) changes due to anthropogenic
activities and how these changes affect PM optical properties is essential for quantifying the
global anthropogenic radiative forcing (IPCC, 2013; Sena et al., 2013). Light absorption
coefficients, $b_{abs}$, and their spectral dependence, commonly referred to as the Ångström
absorption exponent, $\mathring{a}_{abs}$, are needed for accurate interpretation of satellite-retrieved aerosol
optical depth (AOD) for climate modeling. Estimates of the mass absorption efficiency $E_{abs}$ for
PM subcomponents are useful for models to estimate optical effects based on PM composition
and mass concentrations (Laskin et al., 2015).
Organic material that can efficiently absorb radiation in the near-ultraviolet through the
blue end of the visible spectrum, with decreasing absorption efficiency as wavelength increases,
is termed "brown carbon" (BrC) (Pöschl, 2003; Andreae and Gelencsér, 2006; Laskin et al.,
2015). By comparison, black carbon (BC) absorbs light efficiently throughout the visible
spectrum.  Although global climate models have typically treated organic PM as purely
scattering, several studies have shown that brown carbon can contribute substantially to light
absorption by PM, especially in regions affected by biomass burning and urban emissions
(Andreae and Gelencsér, 2006; Ramanathan et al., 2007; Bond et al., 2011; Bahadur et al., 2012;
Ma and Thompson, 2012; Feng et al., 2013). In addition to primary emissions of BrC, secondary
production of BrC can occur from the oxidation of volatile organic compounds (VOCs) present
in biomass smoke (Saleh et al., 2014) and from atmospheric multiphase reactions involving a
wide range of precursor VOCs (Nozière et al., 2007; De Haan et al., 2009; Nguyen et al., 2012;



Lee et al., 2013; Lin et al., 2014; Powelson et al., 2014). The specific sources, chemical
characteristics, and optical properties of BrC remain largely unconstrained.
Biomass burning and urban pollution can affect the concentrations, composition, and
properties of atmospheric PM. In Amazonia, urban pollution is significant downwind of large
cities such as Manaus, Brazil (Kuhn et al., 2010; Martin et al., 2017; Cirino et al., 2018; de Sá et
al., 2018). Martin et al. (2017) reported increased concentrations of particles, nitrogen oxides,
carbon monoxide, and hydroxyl radicals for in-plume compared to out-of-plume conditions
downwind of Manaus. Liu et al. (2016) and de Sá et al. (2017) demonstrated that the Manaus
pollution plume shifted the oxidation pathway of isoprene, thereby significantly affecting gas-
and particle-phase compositions. de Sá et al., 2018 determined that the submicron PM mass
concentration increased by up to three-fold for polluted compared to background conditions
downwind of Manaus during the wet season.
Most biomass burning in Amazonia is related to human activities (Davidson et al., 2012;
Artaxo et al., 2013; Aragão et al., 2014; van Marle et al., 2017). Among the main activities are
the clearing of land and the burning of waste for several agricultural purposes as well as the
burning of wood as fuel (Crutzen and Andreae, 1990; van Marle et al., 2017). Burning events are
most frequent in the period of August through October, corresponding to the dry season (Setzer
and Pereira, 1991; Artaxo et al., 2013; Martin et al., 2016). These activities can affect the
biogeochemical cycles, atmospheric chemistry, precipitation, and climate throughout Amazonia
(Crutzen and Andreae, 1990; Andreae et al., 2004; Lin et al., 2006). $PM_1$ mass concentrations
typically increase by an order of magnitude between the wet and dry seasons in the Amazon,
which has been commonly attributed to the increased biomass burning emissions (Artaxo et al.,
1994; Holben et al., 1996; Martin et al., 2010b; Artaxo et al., 2013, and references therein).



Related increases in $b_{abs}$ by one order of magnitude have also been attributed to biomass burning
(Rizzo et al., 2011; Artaxo et al., 2013; Rizzo et al., 2013). Although black carbon is usually the
main light-absorbing component for atmospheric particles smaller than 1 μm ($PM_1$), absorption
by the organic BrC component of $PM_1$ could also be significant (Rizzo et al., 2013; Wang et al.,
2016; Saturno et al., 2017). Palm et al. (2018) showed that the formation potential of secondary
organic $PM_1$ increased by a factor of 1.7 in the dry season compared to the wet season, although
biomass burning gases were not dominant precursors in either season. An understanding of the
types and optical properties of organic components that may affect $PM_1$ light absorption in the
Amazon and elsewhere is still emerging (Laskin et al., 2015).

The study herein investigates the contributions of biomass burning, urban emissions, and

biogenic emissions to the composition and optical properties of organic $PM_1$ in central Amazonia
during the dry season. Positive-matrix factorization (PMF) of organic mass spectra measured by
an Aerosol Mass Spectrometer (AMS) was used to identify component classes of the organic
$PM_1$. A Fuzzy c-means clustering analysis of pollution indicators was employed to identify
different conditions at the measurement site, as influenced by biomass burning and urban
emissions. Connections are made between the optical properties of organic $PM_1$, including
$b_{abs,BrC}$ and $E_{abs}$, and its component classes. Taken together, these three pieces of analysis allow
for insights into the changes in particle concentration, composition, and optical properties
associated with the influences of biomass burning and urban pollution downwind of Manaus.
**2.  Methodology**
**2.1    Research sites and measurements**

The primary site of this study was called "T3", located 70 km to the west of Manaus,

Brazil, in central Amazonia (Martin et al., 2016; inset of Figure 1a). The pollution plume



primarily passed westerly of Manaus in the dry season and was modeled to intercept the T3 site
about 60% of the time (Martin et al., 2017). Analyses of observational datasets have labeled
pollution episodes for at least 15 to 30% of the time (Thalman et al., 2017; Cirino et al., 2018).
Auxiliary sites "T0a" and "T2" served as references for background and urban-polluted
conditions, respectively, in relation to T3. The T0a site was located at the Amazon Tall Tower
Observatory (Andreae et al., 2015), about 150 km to the northeast of Manaus, and the air masses
were typically upwind of the urban region without the influence of Manaus pollution. The T2 site
was located 8 km to the west of Manaus, directly downwind of the city, and air masses were
therefore typically heavily polluted at this site. During the dry season, the three sites were also
affected by both nearby and long-range transported biomass burning emissions. The study period
from August 15 to October 15, 2014, corresponded to the second Intensive Operating Period
(IOP2) of the GoAmazon2014/5 experiment (Martin et al., 2016).

At the T3 site, mass concentrations of non-refractory $PM_1$ components (organic, sulfate,

ammonium, nitrate, and chloride) were measured by a High-Resolution Time-of-Flight Aerosol
Mass Spectrometer (AMS; DeCarlo et al., 2006; Sueper et al., 2018). A detailed description of
operation was provided in de Sá et al. (2017). In brief, the AMS was deployed inside a
temperature-controlled research container, and ambient data were collected every other 4 min.
Data analysis was performed using *SQUIRREL* (1.56D) and *PIKA* (1.14G) of the AMS software
suite (DeCarlo et al., 2006). The mean composition-dependent collection efficiency was 0.51
(Section S1; Figure S1) (Middlebrook et al., 2012). Organic and inorganic nitrate concentrations
were estimated from the AMS measurements based on the ratio of the signal intensity of $NO_2^+$ to
that of $NO^+$ (Supplementary Material, Section S1, Figure S2) (Fry et al., 2009; Farmer et al.,
2010; Fry et al., 2013). Sulfate measured by the AMS includes contributions from organo-





sulfates (Farmer et al., 2010; Glasius et al., 2018). The oxygen-to-carbon (O:C) and hydrogen-to-
carbon (H:C) ratios of the organic PM$_1$ were calculated following the methods of Canagaratna et
al. (2015).
Several other instruments complemented the AMS measurements. Gas- and particle-
phase semi-volatile tracers were obtained by a Semi-Volatile Thermal Desorption Aerosol Gas
Chromatograph (SV-TAG) (Isaacman-VanWertz et al., 2016; Yee et al., 2018), and VOCs were
obtained by a Proton-Transfer-Reaction Time-of-Flight Mass Spectrometer (PTR-ToF-MS) (Liu
et al., 2016). In addition, measurements of NO$_y$, O$_3$, particle number, and CO concentrations
were employed in the analyses (Martin et al., 2016). Refractory black carbon (rBC)
concentrations were measured by a Single Particle Soot Photometer (SP2). Meteorological
variables, including temperature, relative humidity, and solar irradiance were also measured.
Particle absorption coefficients $b_{abs}(\lambda)$ were obtained by a seven-wavelength aethalometer (370,
430, 470, 520, 565, 700, and 880 nm; Magee Scientific, model AE-31) following the methods
and corrections of Rizzo et al., 2011. Additional measurements of non-refractory particle
composition and concentration from the T0a and T2 sites were made by an Aerosol Chemical
Speciation Monitor (ACSM) at each site (Ng et al., 2011; Andreae et al., 2015; Martin et al.,

2016).

Air-mass backtrajectories were estimated using HYSPLIT4 (Draxler and Hess, 1998).
Simulations started at 100 m above T3 and were calculated for every 12 min up to two days back
in time. Input meteorological data on a grid of 0.5° × 0.5° were obtained from the Global Data
Assimilation System (GDAS). Precipitation along the trajectories was based on data sets of the
S-band radar of the System for Amazon Protection (SIPAM) in Manaus (Machado et al., 2014).





Additional information on the backtrajectory calculations and on the radar were described in de
Sá et al. (2018).

### 2.2 Brown carbon light absorption

The analysis partitioned the total absorption $b_{abs}(\lambda)$ measured by the aethalometer
between BrC and BC contributions, as follows:
$$b_{abs} = b_{abs,BrC} + b_{abs,BC} \qquad (1)$$
The dependence on wavelength was expressed by the absorption Ångström exponent $\mathring{a}_{abs}$, as
follows:
$$\mathring{a}_{abs}(\lambda_1, \lambda_2) = -\frac{\log_{10}[b_{abs}(\lambda_1)/b_{abs}(\lambda_2)]}{\log_{10}(\lambda_1/\lambda_2)} \qquad (2)$$
For the characterization of BrC absorption, the value of at 370 nm was sought. To
calculate $b_{abs,BrC}(370)$, an assumption has to be made about the spectral dependency of BC light
absorption. In this study, $\mathring{a}_{abs,BC}$ was assumed to be wavelength-independent, and $\mathring{a}_{abs,BC}(700,880)$
was calculated for each sample based on $b_{abs}$ at the wavelengths 700 and 880 nm (Eq. 2),
assuming absorption to be insignificant for BrC and dominated by BC in this spectral range.
Calculations of $b_{abs,BrC}(370)$ using alternative treatments to retrieve $\mathring{a}_{abs,BC}$ were also carried out.
These treatments included the assumption that $\mathring{a}_{abs,BC}$ is equal to 1.0 and wavelength-independent
(e.g., Yang et al., 2009), or the assumption that $\mathring{a}_{abs,BC}$ has a spectral dependency itself (Wang et
al., 2016; Saturno et al., 2018a). The results from these different treatments correlated with one
another ($R^2 > 0.9$), and the $b_{abs,BrC}$ estimate used in this study and detailed in the steps below
represented a lower bound among the differing assumptions (Section S4).
For each point in time, $b_{abs,BrC}(370)$ was estimated by the following steps: (1) $b_{abs,BC}(700)$
$= b_{abs}(700)$ and $b_{abs,BC}(880) = b_{abs}(880)$ assuming that $b_{abs,BrC} = 0$ at red wavelengths, (2)
$\mathring{a}_{abs,BC}(700,880)$ was calculated from Equation 2 using $b_{abs,BC}(700)$ and $b_{abs,BC}(880)$, (3)





$b_{abs,BC}(370)$ was calculated from Equation 2, using $b_{abs,BC}(880)$ and $å_{abs,BC}(370,880) =$
$å_{abs,BC}(700,880)$ under the assumption that $å_{abs,BC}$ was independent of wavelength, and finally (4)
$b_{abs,BrC}(370)$ was obtained by Equation 1 using $b_{abs,BC}(370)$ and $b_{abs}(370)$. The value of $b_{abs,BrC}$ at
430 nm was also obtained by the same process. Based on $b_{abs,BC}(370)$ and $b_{abs,BC}(430)$,
$å_{abs}(370,430)$ was estimated. Hereafter, $b_{abs}$ and $b_{abs,BrC}$ refer to 370 nm, and $å_{abs}$ refers to the
range of 370 to 430 nm.

The aethalometer, like other filter-based measurement schemes (e.g., PSAP, TAP, or

MAAP), is prone to artifacts. These artifacts may originate from light scattering by the filter
media itself, the influence of the filter media on the microphysical properties of the collected
particle (e.g., potential change in hygroscopic particle size), and the impact of the multiple
scattered photons on the measured optical extinction (e.g., enhanced particle absorption as
discussed by Nakayama et al., 2010). While several correction schemes have been developed to
address these artifacts, the individual schemes do not approach these problems in the same way,
which may lead to different results among them (Weingartner et al., 2003; Schmid et al., 2006;
Collaud Coen et al., 2010; Rizzo et al., 2011; Ammerlaan et al., 2017). For the present analysis,
the correction scheme used was described by Rizzo et al., 2011.  The potential impact of the
different correction schemes on the analysis interpretation was not examined.
**3. Results and discussion**
**3.1    Contributions of biomass burning and urban emissions to fine-mode PM**
**3.1.1    Comparison of PM concentration and composition across sites**

A comparison between the T3 site and the upwind sites can provide a first-order estimate

of the effects of Manaus urban pollution on $PM_1$ concentration and composition (de Sá et al.,
2018). During the dry season of 2014, organic compounds dominated the composition at T3,



contributing $83 \pm 6\%$ (mean ± one standard deviation) of the non-refractory $PM_1$ (NR-$PM_1$),
followed by sulfate ($11 \pm 5\%$) (Figure 1a). Mean NR-$PM_1$ mass concentrations and relative
compositions at T3 and at T0a and T2 are represented in Figure 1b for comparison. Organic
material consistently constituted 80% to 85% of NR-$PM_1$ across all three sites. By comparison,
the contribution of organic material to NR-$PM_1$ typically ranged from 70 to 80% during the wet
season (de Sá et al., 2018).
The NR-$PM_1$ mass concentrations across the three sites differed slightly (Figure 1b, top
panel). The mean concentration at the T0a site upwind of Manaus was 10.5 µg m$^{-3}$. The mean
concentrations at the T2 site just downwind of Manaus and at the T3 site further downwind were
12.5 µg m$^{-3}$ and 12.2 µg m$^{-3}$, respectively, representing an increase of about 20% relative to the
upwind site. By comparison, increases of 200 to 300% relative to the upwind site were observed
during the wet season (de Sá et al., 2018). In absolute mass concentration, however, the
difference between upwind and downwind sites of 1 to 2 µg m$^{-3}$ was similar between seasons,
suggesting contributions from urban pollution in the same order of magnitude in both seasons.
The larger percent increase for the wet season is explained by background concentrations of 1 µg
m$^{-3}$ which are an order of magnitude lower compared to the dry season.
The time series of organic and sulfate mass concentrations across the three sites were
highly correlated across the two months (Figure 2). The T0a and T3 sites were separated by 215
km. This result shows that sources and processes of $PM_1$ production at a regional scale were
important during the dry season. The figure also shows that for timescales of less than a day the
sites were less correlated. The large spikes in organic mass concentrations observed at T3 but
generally smaller at T2 and absent at T0a could be explained by episodic fires along the
Solimões River, especially during nighttime (Figure 3).



In addition to the widespread and frequent occurrence of fires in the Amazon basin
during the dry season (Figure 3), meteorological conditions may also favor a regional reach of
events (Section S3). For example, high organic concentrations were observed during the period
of August 17 to 23. During that week, widespread biomass burning activity in the basin (beyond
the scale of Figure 3) in conjunction with a lack of precipitation events, clear skies, and
temperatures of 35 °C during daytime allowed for intense photochemical activity and buildup of
$PM_1$. There appeared to be an offset in $PM_1$ concentrations by 1 day between T0a and T3 during
that time, which would be consistent with the transport across 215 km from T0a to T3 for typical
easterlies averaging 3 m s$^{-1}$ over the course of a day. In short, a combination of regional-scale
biomass burning activity and meteorological conditions greatly influenced the mass
concentration of $PM_1$ at the three sites.
The diel variability of organic and sulfate mass concentrations for the three sites is shown
in Figure 4. Organic mass concentrations were slightly higher at the T2 and T3 sites compared to
the T0a site, as expected. The variability was larger at the T2 and T3 sites, especially so at night.
These two sites are closer to populated areas along the path of the Solimões River and thus are
also closer to local biomass burning sources. Activities include burning of crops and trash in
houses and farms as well burning of wood in brick kilns (Martin et al., 2016; Cirino et al., 2018).
Stagnant air and a shallow boundary layer during the night might explain how variable biomass
burning emissions lead to larger organic mass concentrations and variability at night compared to
the day.
The influence of anthropogenic emissions on daytime chemistry is apparent in the diel
trends of the sulfate mass concentrations. Sulfate concentrations had low variability throughout
the day at T0a, indicating a prevalence of diffuse regional sources that had variations dampened





after many hours or days of transport. Possible sources include the atmospheric oxidation of
biogenic emissions (DMS, H$_2$S) from the upwind forest and ocean, as well as long-range
transport of fossil fuel combustion emissions from cities in northeastern Brazil and of biomass
burning and volcanic emissions from Africa (Andreae et al., 1990; Martin et al., 2010a; Saturno
et al., 2018b.) Biomass burning can be an important source of sulfate and its precursors (Andreae
and Merlet, 2001; Fiedler et al., 2011). For the T2 and T3 sites, sulfate concentrations increased
in the morning hours and peaked in the afternoon. The Manaus sulfate source consists of the
burning of heavy fuel oil for electricity production, refinery operations, and more diffuse traffic
sources, and these emissions reach the T3 site in the afternoon, when OH levels are also the
highest (de Sá et al., 2017). In addition, biomass burning emissions around T2 and T3 might also
have contributed to the increase in sulfate concentrations during the afternoons.
**3.1.2  Comparison of PM concentration and composition across clusters for the T3 site**
A second approach to investigate the changes in concentrations and compositions of the
PM with pollution influences employed a combination of positive-matrix factorization (PMF)
and Fuzzy c-means (FCM) clustering. The PMF analysis was applied to the organic mass spectra
to separate the organic PM$_1$ into representative component classes (section 3.1.2.1). The FCM
clustering algorithm was applied to auxiliary measurements to identify times of urban and
biomass burning influences at the T3 site (section 3.1.2.2). The results of the FCM analysis were
crossed with the findings of the PMF analysis for further insights into pollution-related
variability of PM concentration and composition (section 3.1.2.3).
**3.1.2.1 Classification of organic PM by positive-matrix factorization**
The organic mass spectra recorded by the AMS at the T3 site were analyzed by PMF
(Ulbrich et al., 2009). Details and diagnostics of the PMF analysis are presented in the



Supplementary Material (Section S1). Following the nomenclature used in de Sá et al. (2018),
"mass spectrum" and "mass concentration" refer to the direct AMS measurements, while "factor
profile" and "factor loading" are their counterpart  mathematical products obtained from the
PMF analysis. A six-factor solution was obtained, and the factor profiles, diel trends of the factor
loadings, and the time series of the factor loadings and other related measurements are plotted in
Figure 5. The correlations of factor loadings with co-located measurements of gas- and particle-
phase species are presented in Figure 6.

The factors were interpreted considering the mass spectral characteristics of the factor

profiles and the correlations between factor loading and mass concentrations of co-located
measurements. Three resolved factors interpreted as secondary production and processing
closely matched the counterpart profiles of the wet season ($R \geq 0.99$; Table 1) (de Sá et al.,
2018). These three factors consisted of a more-oxidized oxygenated factor ("MO-OOA"), a less-
oxidized oxygenated factor ("LO-OOA"), and an isoprene epoxydiols-derived factor ("IEPOX-
SOA"). Temporal correlations with external tracers and oxidation characteristics were also
similar to those of the wet season, corresponding to IOP1 (Figure 6; Table 1; de Sá et al., 2018).
Although a hydrocarbon-like factor ("HOA") was analogous to its counterpart in IOP1 ($R =$
0.94), it also had characteristics of an IOP1 anthropogenic-dominated factor ("ADOA") tied to
other urban sources including cooking. The HOA factor of IOP2 therefore represented a mix of
the HOA and ADOA factors of IOP1, which could not be separated by PMF in IOP2 due to their
lower relative contributions. The interpretation of the HOA, IEPOX-SOA, LO-OOA, and MO-
OOA factors follows that of IOP1, as presented in de Sá et al. (2018). The following discussion
focuses on the two biomass burning factors of IOP2.





A less-oxidized factor ("LO-BBOA") and a more-oxidized factor ("MO-BBOA") were

resolved for IOP2. For IOP1, a single "BBOA" factor was resolved, and it accounted for 9% of
the organic $PM_1$ mass concentration. For IOP2, there were enough differences in mass spectral
features and temporal contributions, as well as larger overall contributions of biomass burning,
that the PMF analysis identified two different factors. The MO-BBOA and LO-BBOA factors
respectively accounted for 18% and 12% of the mean organic $PM_1$ mass concentration.
Therefore, the relative contribution of biomass burning to organic $PM_1$ during the dry season was
at least a factor of three higher compared to the wet season (a more detailed discussion is
presented at the end of this section).

The LO-BBOA and MO-BBOA factor profiles had a distinct peak at nominal $m/z$ 60

($C_2H_4O_2^+$) (Figure 5a). The fractional intensity $f_{60}$ at $m/z$ 60 was larger for LO-BBOA (0.051)
than for MO-BBOA (0.013). A peak at $m/z$ 73 ($C_3H_5O_2^+$) was also present in both profiles,
although its intensity was three to four times smaller than that at $m/z$ 60. The peaks at $m/z$ 60 and
$m/z$ 73 are attributed to fragments of levoglucosan and other anhydrous sugars that are produced
by the pyrolysis of biomass (Schneider et al., 2006; Cubison et al., 2011). Accordingly, the
loadings of both factors correlated with the concentrations of several biomass-burning tracers in
the particle phase, including levoglucosan, vanillin, 4-nitrocatechol, syringol, mannosan,
syringaldehyde, sinapaldehyde, and long-chain alkanoic acids ($C_{20}$, $C_{22}$, $C_{24}$) and of tracers in the
gas phase (acetonitrile) (Figure 6). The loadings also correlated with less-specific tracers,
including CO concentration and particle number concentration. The Pearson-$R$ correlations were
typically higher for the LO-BBOA factor than for the MO-BBOA factor.

The LO-BBOA profile had the greatest ratio of signal intensity of the $C_2H_3O^+$ ion ($m/z$

43) to that of the $CO_2^+$ ion ($m/z$ 44) compared to all other factors (Figure 5a). In comparison, the





MO-BBOA profile had a high intensity for the $CO_2^+$ ion and a low intensity for the $C_2H_3O^+$ ion.
The MO-BBOA and LO-BBOA factors had O:C ratios of $0.70 \pm 0.07$ and $0.53 \pm 0.04$,
respectively. In addition, the LO-BBOA factor loading had higher correlation  with the estimated
inorganic nitrate concentrations than with the total nitrate concentrations whereas the MO-
BBOA factor did not (Figure 6; Supplementary Material, Section S1 describes the nitrate
estimates). Taken together, these results point to a less-oxidized, higher-volatility character of
the LO-BBOA factor and a more-oxidized, lower-volatility character of the MO-BBOA factor,
both with biomass-burning characteristics (Jimenez et al., 2009; Cubison et al., 2011; Gilardoni
et al., 2016; Zhou et al., 2017).

The extent of the biomass burning influence and atmospheric oxidation on the

composition of organic $PM_1$ can be visualized in a scatter plot of $f_{44}$ and $f_{60}$ (Figure 7a) (Cubison
et al., 2011). A background $f_{60}$ value of $0.3\% \pm 0.06\%$ (vertical black dashed line) indicates a
threshold for negligible or completely oxidized biomass-burning $PM_1$. Points in the lower right
of the $f_{44}$-$f_{60}$ representation usually characterize $PM_1$ tied to recent biomass burning emissions.
For IOP1 (blue markers), all points lie on or close to the background value suggested by Cubison
et al. (2011), indicating the absence of a strong influence from biomass burning. During the wet
season, biomass burning was limited to local sources or to sources far enough away such as
Africa that the $PM_1$ was extensively oxidized by arrival in central Amazonia (de Sá et al., 2018).
For IOP2 (red markers), the $f_{60}$ values are greater for most observations, showing that for most
times T3 was influenced to some extent by biomass burning (see Section 3.1.2.3). This finding is
in line with the widespread occurrence of fires during the dry season (Figure 3). As suggested by
the robust trend in Figure 7a, the $f_{44}$ value increases and the $f_{60}$ value decreases from the bottom
right to the upper left as the organic $PM_1$ emitted by biomass burning is oxidized in the



atmosphere. The $f_{60}$ and $f_{44}$ values of the LO-BBOA and MO-BBOA profiles, plotted as
diamonds, lie on the linear trend.

The LO-BBOA factor of high $f_{60}/f_{44}$ and low O:C thus appears associated with primary

$PM_1$ emitted by biomass burning. The MO-BBOA factor, characterized by low $f_{60}/f_{44}$ and high
O:C, may represent a combination of primary $PM_1$ of higher oxygen content as well as secondary
$PM_1$ tied to biomass burning in its early stages of atmospheric processing (Cubison et al., 2011;
Gilardoni et al., 2016). These secondary pathways could include (i) the heterogeneous oxidation
of primary $PM_1$, such as that represented by the LO-BBOA factor, and (ii) the oxidation of gas-
phase biomass-burning emissions or of species evaporated from primary $PM_1$, followed by the
condensation of the gas-phase products onto the $PM_1$.

The LO-BBOA and MO-BBOA factor loadings had greater magnitude and variability at

night compared to during day (Figure 5b). Their summed loading, represented as "BBOA$_T$",
accounted for 40% and 13% of the organic $PM_1$ during night and day, respectively. Overall, they
accounted for 30% of the organic $PM_1$. This result reflects the importance of fire activity during
all times of day and during the entirety of IOP2 (Figure 3). The surface concentrations were
lower during the day because biomass burning emissions are diluted with the development of the
planetary boundary layer (PBL) and with the increased wind speeds as compared to the stagnant
air and shallower PBL at night. The occurrence of significant dilution indicates that the emission
sources were at least in part within a day of transport, meaning a distance on the order of a few
hundred kilometers. The fractional contribution of the MO-BBOA factor to BBOA$_T$ shifted from
0.7 to 0.5 from day to nigh, while that of LO-BBOA correspondingly shifted from 0.3 to 0.5
(Figure 7b). This result is consistent with an additional secondary contribution to the MO-BBOA





loading during daytime, including from LO-BBOA oxidation and possibly tied to photochemical
processing, on top of a primary source from biomass burning.

Although the footprint of biomass burning is geographically more widespread throughout

the basin compared to the urban footprint of nearby Manaus, fire incidence and large-scale
emissions have historically concentrated in a region known as the arc of deforestation along the
southern rim of the forest (Fuzzi et al., 2007; Artaxo et al., 2013). Several campaigns have
focused on the effects of biomass burning during the dry season at locations that are highly
affected by fires, usually in the states of Rondônia or Mato Grosso, within the arc of
deforestation (SCAR-B, Kaufman et al., 1998; LBA-SMOCC, Fuzzi et al., 2007; LBA-
EUSTACH, Andreae et al., 2002; TROFEE, Yokelson et al., 2007; SAMBBA, Morgan et al.,
2013). At a ground site in Porto Velho, Rondônia, a PMF analysis of ACSM data showed that
70% of the organic $PM_1$ could be attributed to biomass burning (Brito et al., 2014). Compared to
the present study, in which at least 30% of the organic $PM_1$ can be directly attributed to biomass
burning, the contributions of fires to $PM_1$ in the arc of deforestation region are considerably
larger.

The combined contribution of 30% by MO-BBOA and LO-BBOA at T3 represents a

lower bound of biomass burning influence because more-oxidized material from biomass
burning could be accounted for by the MO-OOA factor. In the limiting assumption that all MO-
OOA loadings originated from BBOA loadings, an upper limit of 50% can be established for the
mean contribution of biomass burning to organic $PM_1$ concentrations at T3. Considering that all
organic $PM_1$ components have been observed to age into MO-OOA at similar rates (Jimenez et
al., 2009), a more likely estimate of 38% can be derived by assuming that all factors contribute to
MO-OOA proportionally to their ambient concentrations.





An important implication of these results, together with those of the wet season, is that
although PM$_1$ concentrations increase on average by a factor of 8.5 between seasons, not all of
the increase is due to biomass burning, which has been a common assumption in previous studies
(Artaxo et al., 1994; Holben et al., 1996; Echalar et al., 1998; Maenhaut et al., 1999; Andreae et
al., 2002; Artaxo et al., 2002; Mace et al., 2003; Martin et al., 2010b; Artaxo et al., 2013; Rizzo
et al., 2013; Brito et al., 2014; Pöhlker et al., 2016). In absolute mass concentrations, the
contribution from biomass burning increased from 0.12 µg m$^{-3}$ in the wet season to 3.4 µg m$^{-3}$ in
the dry season, which represents a 30-fold increase. This result corresponds to a change in
percentage contribution to organic PM$_1$ from 9% to 30% (not counting with the mass presumably
present in MO-OOA). Nevertheless, the contribution from secondary biogenic sources (and their
anthropogenically affected processes), as represented by the LO-OOA and IEPOX-SOA factors,
also increased by around 8-fold from 0.6 µg m$^{-3}$ to 4.8 µg m$^{-3}$. In absolute terms, this mass
increase (of 4.2 µg m$^{-3}$) is comparable to the one associated with biomass burning (3.3 µg m$^{-3}$).
Because the 8-fold mass increase of LO-OOA and IEPOX-SOA was similar to the 8.5-fold
increase in total organic PM$_1$, these factors show a similar mass percentage contribution of 42%
to organic PM$_1$ for both seasons. The MO-OOA factor loadings increased by 6-fold from 0.4 µg
m$^{-3}$ to 2.3 µg m$^{-3}$. Because this relative increase was smaller than that of the total organic PM$_1$,
the MO-OOA factor had a decrease from 30% to 20% of contribution to organic PM$_1$.  The
contribution from urban sources, as represented by the HOA and ADOA factors, increased by
three-fold between seasons, from 0.24 µg m$^{-3}$ to 0.76 µg m$^{-3}$, representing a decrease in mass
percentage contribution from 18% to 7%.
Therefore, reasons other than increased biomass burning in the dry season must have
played a role in increasing organic PM$_1$ concentrations. One aspect is that BVOC emissions are



typically higher in the dry season (Yáñez-Serrano et al., 2015; Alves et al., 2016), which might
partly explain the increases in LO-OOA, IEPOX-SOA, and MO-OOA factors. In addition, the
directly-measured biogenic (total) secondary organic $PM_1$ formation potential of ambient air
increased by a factor of 2.4 (1.7) between seasons (Palm et al., 2018). Increased organic mass
available for partitioning may account for another factor of 2 (Palm et al., 2018). As a
consequence of increased $PM_1$ mass concentrations, the lifetime of semi-volatile gases may also
be increased, since lifetime against dry deposition is much larger for particles than for gases
(Knote et al., 2015). Increased oxidant levels during the dry season could also be a contributing
factor (Rummel et al., 2007; Artaxo et al., 2013; Andreae et al., 2015; Yáñez-Serrano et al.,
2015; Fuentes et al., 2016). Importantly, the mass concentrations of sulfate and ammonium also
increased by six-fold between seasons (Figure S10), indicating that atmospheric physical
processes governing particle mass concentrations possibly played an important role. In this
context, reduced wet deposition due to reduced convection in the dry season may be another
appreciable contributor to the organic $PM_1$ increases (Machado et al., 2004; Nunes et al., 2016
Chakraborty et al., 2018).
**3.1.2.2 Cluster Analysis**

The time series of the afternoon concentrations of particle number, $NO_y$, ozone, rBC,

carbon monoxide, and sulfate were analyzed by Fuzzy c-means clustering at the time resolution
of the AMS measurements. The algorithm attributed degrees of cluster membership to each data
point based on similarity in the sets of input concentrations (Section S2). The scope was
restricted to data sets for which ten-hour air mass back trajectories did not intersect precipitation.
The scope also excluded data sets tied to the lowest 10% of solar irradiance averaged over the
previous 4 h at T3 (Supplementary Material, Section S2). This approach aimed to capture fair-



weather conditions and thereby minimize the role of otherwise confounding processes that
influence mass concentrations, such as boundary layer dynamics and wet deposition.

Three clusters, labeled "baseline," "event," and "urban," were identified based on a

combination of minimization of the FCM objective function and an assessment of
meaningfulness of the resolved set of clusters. Illustrative examples of the obtained degrees of
membership (0 to 1) are plotted in Figure 8a for several time windows. The concentrations of the
input and additional species are plotted in Figures 8b and 8c. The PMF results of section 3.1.2.1
are plotted for comparison in Figure 8d. Air-mass backtrajectories are plotted in Figure 9 for
time windows predominantly associated with only one cluster.

All three clusters reflected, albeit to different degrees, some influence of biomass

burning. For the wet season, de Sá et al. (2018) identified clusters representing background
conditions, which were characterized by low concentrations of particle number, $NO_y$, and $O_3$. For
the dry season, no similar cluster was identified. As shown in Figure 3, there were fires in the
region at all times (cf. Martin et al., 2017).

The baseline cluster had the lowest concentrations of pollutant indicators, representing

influences of far-field biomass burning on top of natural (i.e., biogenic) emissions and
atmospheric processing. The cluster centroid corresponded to 1.3 ppb $NO_y$, 30 ppb ozone, and
2000 particles $cm^{-3}$ (Table S1). Results for August 27, August 28, and September 9 illustrate
these lower concentrations compared to the other days (Figure 8). The backtrajectories
associated with the baseline cluster did not intersect the urban area of Manaus, especially the
southern region of  presumed higher emissions (Figure 9a; de Sá et al., 2018).

The event cluster referred to conditions of increased influence from biomass burning and

long-range transport of volcanic emissions from Africa. The cluster corresponded to a 10-day





period from Sep 22 to Oct 1 in which biomass burning intensified in the surroundings of T3 as
well as more broadly in the Amazon basin (Figures 3f and 3g). Coincidentally, plumes carrying
emissions from the Nyamuragira-Nyiragongo volcanoes in Africa were also observed to reach
central Amazonia during that time period , as demonstrated by Saturno et al. (2018b). This
cluster was characterized by higher concentrations of all species in relation to the baseline cluster
(Table S1). In particular, the sulfate concentrations (2.3 µg m$^{-3}$ at the centroid) were the highest
among the three clusters. Results for September 23, September 27, and September 28 illustrate
these findings for T3, with sulfate concentrations reaching 4 µg m$^{-3}$ (Figure 8). This trend in
sulfate concentrations was consistent across all three sites (Figure 2). The backtrajectories
associated with the event cluster were variable, passing to the north, directly over, and to the
south of Manaus, although always with an east component (Figure 9b). The long-range transport
and increased regional fire count during the event period thus appeared more important in
defining this cluster than did the directions of the backtrajectories in a smaller scale, making
Manaus emissions of secondary importance.

The urban cluster had the highest centroid concentrations of NO$_y$ (2.6 ppb), ozone (56.4

ppb), and particle number (4600 cm$^{-3}$) among the three clusters (Table S1). It represented
conditions for which both biomass burning and urban emissions were relevant, and these
emissions may have interacted before reaching the T3 site. The results for August 24, September
11, September 14, and October 8 illustrate the high pollutant concentrations (Figure 8). The
backtrajectories associated with the urban cluster consistently passed over Manaus and, more
specifically, over the southern region where human activities were more concentrated (Figure
9c).



### 3.1.2.3 Comparison of PM$_1$ composition among clusters


Species mass concentrations and PMF factor loadings associated with the cluster

centroids were determined (Section S2). The resulting organic, sulfate, ammonium, nitrate, and
chloride mass concentrations associated with each cluster are represented in Figure 10a. The
PMF factor loadings associated with each cluster are likewise represented in Figure 10b.

The summed NR-PM$_1$ mass concentrations for the centroids of the event and urban

clusters were both 12.3 μg m$^{-3}$. This concentration was 33% higher than that representing the
baseline cluster (9.2 μg m$^{-3}$). This result thus agrees with that based on direct comparison of PM$_1$
mass concentrations between the T3 and the T0a sites (Section 3.1.1). Therefore, the overall
effect of Manaus pollution was to add 1 to 3 μg m$^{-3}$ on top of the upwind concentrations.
Increases in the organic mass concentration dominated the overall increase in PM$_1$ mass
concentration because organic species dominated the composition for all three clusters. The
increases in organic mass concentration for the event and urban clusters relative to the baseline
cluster were 26% and 33%, respectively (Figure 10a).

Sulfate concentrations also increased relative to the baseline cluster, corresponding to

65% for the event cluster and 31% for the urban cluster. This result indicates that strong biomass
burning emissions reaching areas downwind of Manaus as well as long-range transport of
volcanic emissions from as far away as Africa may increase sulfate concentrations in those areas
beyond the sulfate values driven by the anthropogenic activities in the city.  In other words, ,
there were several other in-basin as well as out-of-basin sources of sulfate  besides Manaus that
could sustain relatively high sulfate concentrations (Chen et al., 2009; de Sá et al., 2017; Saturno
et al., 2018b).



The relationship between clusters and PMF factors is represented in Figure 10b. All three
clusters were associated with an organic $PM_1$ composition dominated by secondary production.
The baseline cluster was largely dominated by the LO-OOA factor (40%). By comparison, the
event cluster had significant increases in the LO-BBOA, MO-BBOA, and IEPOX-SOA factor
loadings. The increase in LO-BBOA and MO-BBOA loadings (40%) can be associated with the
increased contributions of primary and secondary particle components from biomass burning,
respectively. The LO-BBOA factor had the highest loading (0.5 $\mu g\ m^{-3}$) for the event cluster,
consistent with the high incidence of fires during the period represented by this cluster. The
increase of 65% in IEPOX-SOA loading can be explained by the disproportionally higher
increase of 65% in the sulfate concentration (which favors higher IEPOX-SOA loadings),
accompanied by the relatively moderate increase of 34% in $NO_y$ concentration, (which
suppresses IEPOX-SOA loadings), leading to a net increase in IEPOX-SOA loadings (Table S1;
de Sá et al., 2017).
The composition of the organic $PM_1$ associated with the urban cluster differed from that
of the two other clusters, as indicated by the factor contributions (Figure 10). Compared to the
baseline cluster, the loadings of all factors except IEPOX-SOA increased. An increase in HOA
loading is consistent with emissions in the city, including from vehicles and power plants. An
increase in the loadings associated with secondary processes, as represented by the MO-OOA,
LO-OOA, and MO-BBOA factors, can be explained by the accelerated oxidation cycle in the
plume. In brief, an increase in the concentrations of both precursors and oxidants provided by
urban emissions accelerates the production of secondary $PM_1$ and thereby increases the $PM_1$
concentrations downwind of the city (Martin et al., 2017; de Sá et al., 2018).





The similarity in IEPOX-SOA factor loading for the baseline and the urban clusters may
be explained by the following aspects. First, the lifetime of IEPOX-derived PM in the boundary
layer is thought to be around 2 weeks (Hu et al., 2016). Therefore, a substantial fraction of this
component observed at T3 will be formed upwind of the Manaus plume. Second, favored
conditions for IEPOX production and uptake are low NO concentrations (i.e., HO$_2$-dominant
pathway for the ISOPOO radical) and high sulfate concentrations (de Sá et al., 2017). Sulfate
concentrations increased by 31%, and NO$_y$ concentrations, used as an indicator for exposure of
the airmass to NO concentrations, increased by 100% for the urban compared to the baseline
cluster. These two changes work against one another with respect to IEPOX production and
uptake. For the wet season, de Sá et al. (2017) reported that the IEPOX-SOA factor loading was
more sensitive to changes in NO$_y$ concentration for 1 ppb and less. By comparison, NO$_y$
concentrations in the dry season were consistently greater than this value. Due to this lower
sensitivity, large increases in NO$_y$ may not be tied to large decreases in IEPOX-SOA factor
loading in the dry season. In sum, the opposite roles of sulfate and NO$_y$ concentrations can
explain the net zero change in IEPOX-SOA factor loadings between baseline and urban clusters.
Because all of the loadings for other factors increased, the fractional loading of IEPOX-SOA
decreased from 26% to 15%.
**3.2    Contributions of biomass burning and urban emissions to brown carbon**
**3.2.1    Brown carbon light absorption**
The diel trends of $b_{abs}$, $b_{abs,BrC}$, $b_{abs,BrC}/b_{abs}$, and $å_{abs}$ are shown in Figure 11. Both $b_{abs}$ and
$b_{abs,BrC}$ were larger and had greater variability at night compared to day. The variability of the
fractional contribution of BrC to the total absorption, represented by $b_{abs,BrC}/b_{abs}$, was smaller
than the variability of its components $b_{abs}$ and $b_{abs,BrC}$ (i.e., Figure 11c compared to Figures 11a-





b). The absorptive contributions of BC and BrC thus co-varied to some extent, suggesting a
partial overlap in sources, which is consistent with previous studies (Collier et al., 2016; Jen et
al., 2018). Furthermore, the fractional contribution $b_{abs,BrC}/b_{abs}$ increased from 0.2 in the day to
0.4 at night. Compared to the diel trends of the six PMF factor loadings, the diel trends of the
absorption properties were most similar to those of the MO-BBOA, LO-BBOA, and HOA
factors (Figure 5b).

Figure 12 illustrates connections between $b_{abs,BrC}$ and the organic $PM_1$ chemical

composition. Brown-carbon light absorption decreases for increases in the O:C ratio (Figure
12a). Conversely, light absorption increases for decreases in the concentration of nitrogen-
containing species, as represented by the $C_xH_yO_zN_p^+$ family (Figure 12b). In addition, light
absorption increases as the fractional contribution of the $C_xH_yO_zN_p^+$ family to organic $PM_1$
increases and that of the $C_xH_yO_z^+$ family decreases (Figure S14). The diel trends of Figure 11 and
the O:C ratios of Figure 12a support an association of brown-carbon light absorption with HOA
and LO-BBOA factor loadings. These factors had the lowest O:C values (Table 1), and they are
associated with recent urban and biomass burning emissions, which are typically important
sources of brown carbon (Laskin et al., 2015, and references therein).

The decrease in $b_{abs,BrC}$ as O:C increases suggests that the atmospheric processing of

organic material bleaches the BrC components under the conditions of central Amazonia. This
behavior has been observed in several laboratory studies: BrC species and thus their optical
properties can be modified through atmospheric processing, which may involve reactions at the
gas-particle interface, reactions in the aqueous phase of particle and cloud droplets, and
photolysis driven by sunlight (Laskin et al., 2015; Zhao et al., 2015; Sumlin et al., 2017;Lee et
al., 2014; Romonosky et al., 2015). In addition, Saleh et al. (2014) provided evidence that both



primary and secondary material from biomass burning may absorb light, and that the secondary
component may be less absorptive than the primary component in the visible spectral range. Lin
et al. (2016) found that the absorbance at 300 nm by biomass burning particles decayed with a
half-life of approximately 16 h against photolysis under typical atmospheric conditions. Forrister
et al. (2015) followed plumes from wildfires onboard an aircraft during the 2013 NASA
SEAC4RS mission over the continental USA and estimated a half-life of 9 to 15 h for the decay
of BrC light absorption in the plumes.

An important contribution of nitrogen-containing organic molecules to $b_{abs,BrC}$ is

suggested by the relationship in Figure 12b. The percent contribution of the $C_xH_yO_zN_p^+$ family to
each PMF factor profile is listed in Table 2  and is highest for the HOA and LO-BBOA factors.
The correlations of factor loadings with the $C_xH_yO_zN_p^+$ mass concentrations as well as with the
$b_{abs,BrC}$ values are highest for these two factors ($R > 0.8$ and $R > 0.6$, respectively) (Table 2). The
correlations of the MO-BBOA factor loading with these two parameters are lower but still
significant. By comparison, the corresponding correlations for the IEPOX-SOA, LO-OOA, and
MO-OOA factor loadings are all lower than 0.5. These results further support that the HOA and
LO-BBOA factors to a larger extent and the MO-BBOA factor to a lesser extent were tightly
associated with nitrogen-containing, light-absorbing organic molecules.

In contrast to the $C_xH_yO_zN_p^+$ family, the correlations between PMF factor loadings and

mass concentrations of organic nitrates are low ($R < 0.4$, Table 2; Figure S12). For the HOA,
LO-OOA, and MO-OOA factors associated with BrC light absorption, the correlations are small
($R < 0.25$).  The implication could be that the $C_xH_yO_zN_p^+$ family is closely tied to $PM_1$
constituted by reduced nitrogen compounds and nitrogen-aromatic compounds. By comparison,





organic nitrates are more strongly tied to photochemical production of secondary PM$_1$ and
represent more oxidized forms of nitrogen, including in aliphatic molecules.

Several studies have suggested that nitrogen-containing organic molecules are important

absorbers in organic PM (Sun et al., 2007; Lin et al., 2016). Claeys et al. (2012) characterized
humic-like substances (HULIS) present in PM collected during the biomass burning season in
Amazonia and identified nitro-aromatic catechols and aromatic carboxylic acids among the main
constituents. Nitrophenol derivatives have been identified as major BrC components in several
other urban and rural locations worldwide (Kitanovski et al., 2012; Desyaterik et al., 2013; Kahnt
et al., 2013; Mohr et al., 2013). Importantly, Lin et al., 2016 further verified that compounds that
are usually interpreted as secondary, such as nitro-phenols and derivatives, can be produced in
the heat-laden, VOC-rich, high-NOx conditions of the biomass burning process, being
subsequently emitted as primary material. Furthermore, Yee et al. (2013) observed the quick
conversion of guaiacol and syringol to nitro-guaiacol and nitro-syringol, respectively, in the
presence of HONO even without heat or photo-oxidation. It is possible that BrC from other
combustion sources could have similar characteristics based on this reasoning, helping to explain
the association found in this study between BrC absorption and the LO-BBOA and HOA factors.
Regarding the further atmospheric processing of these nitrogen-containing organic compounds,
laboratory studies have shown  that hydroxy radical oxidation of nitro-aromatic species in
aqueous solutions leads to fragmentation into smaller organic acids (e.g., oxalic, glycolic,
malonic, and isocyanic) or, in general, reduce the size of the conjugated molecular systems,
leading to a decrease in light absorption at visible wavelengths (Sumlin et al., 2017; Hems and
Abbatt, 2018). These findings may help to explain the bleaching of BrC as the material becomes
more oxidized. In the context of the PMF factors, these smaller later-generation products may



then be associated with the MO-OOA factor or may partition to the gas phase depending on their
volatility.

Scatter plots of $\mathring{a}_{abs}$ against markers of biomass burning are shown in Figure 13. The

Pearson-$R$ correlations against $\log_{10}(f_{60}/f_{44})$ and (BBOA$_T$/organic PM$_1$) are 0.87 and 0.75,
respectively. The $f_{60}/f_{44}$ ratio is a tracer for the influence of fresh biomass burning, and an
association of $\mathring{a}_{abs}$ and with this quantity was also reported for Boulder, Colorado, USA (Lack et
al., 2013). These relationships could be useful parameterizations to estimate $\mathring{a}_{abs}$ when optical
measurements are not available but AMS / ACSM measurements are, at least during times of
biomass burning influence in central Amazonia. Worldwide, observed values of $\mathring{a}_{abs}$ range from
<2 to 11 for particles tied to biomass burning (Chakrabarty et al., 2010; Saleh et al., 2014). The
value of $\mathring{a}_{abs}$ reached 6 for the highest observed values of ($f_{60}/f_{44}$). It approached 1.0 in the limit of
$f_{60}/f_{44} < 0.02$, which indicates little influence of proximate biomass burning (Figure 13a; cf.
upper left of Figure 7a). Further observations elsewhere in the Amazon and on other regions are
needed before the parameterizations suggested by Figure 13 between $\mathring{a}_{abs}$ and markers of
biomass burning can be generalized with confidence.
**3.2.2  Contribution of organic PM components to BrC absorption**

Herein, advantage is taken of the representation of the organic PM in its subcomponents

provided by the PMF factors to estimate a mass absorption efficiency for each of them. The
absorption coefficient is the sum of the absorption coefficient of the $n$ parts of the organic PM
("Org"):

$$b_{abs,BrC} = b_{abs,Org_1} + b_{abs,Org_2} + \ldots + b_{abs,Org_n} \tag{3}$$


The treatment assumes the absence of cross-interactions among the parts and holds for a single
wavelength. The absorption coefficient $b_{abs,i}$ of part $i$ is defined as follows:





$$b_{\text{abs},i} = E_{\text{abs},i} \times C_i \qquad (4)$$

where $E_{\text{abs,i}}$ is the mass absorption efficiency and $C_i$ is the mass concentration of part $i$. Based on
equations 3 and 4, the following model was constructed for $b_{\text{abs,BrC}}$ by using the PMF factor
loadings as a proxy for the mass concentrations of organic $PM_1$ components:
$$b_{\text{abs,BrC}} = E_{\text{abs,MO-OOA}}\, G_{\text{MO-OOA}} + E_{\text{abs,LO-OOA}}\, G_{\text{LO-OOA}} + E_{\text{abs,IEPOX-SOA}} G_{\text{IEPOX-SOA}} +$$
$$+ E_{\text{abs,MO-BBOA}}\, G_{\text{MO-BBOA}} + E_{\text{abs,LO-BBOA}}\, G_{\text{LO-BBOA}} + E_{\text{abs,HOA}}\, G_{\text{HOA}} + B \qquad (5)$$

where $G_i$ correspond to loadings of factor $i$, and the unknowns are the mass absorption
efficiencies $E_{\text{abs,i}}$ associated with each PMF factor. An intercept $B$ was added to account for the
variability not explained by the PMF factors. Other studies have also made use of multivariate
linear regression to retrieve mass absorption efficiencies (Hand and Malm, 2007;Washenfelder et
al., 2015).
Estimates of $E_{\text{abs,i}}$ were obtained using a constrained linear least-squares algorithm
applied to Eq. 5, where the inputs were the observed $b_{\text{abs,BrC}}$ and the factor loadings for each
point in time during IOP2. The input data are represented in the scatter plots of $G_i$ against $b_{\text{abs,BrC}}$
shown in Figures 14a to 14f. A non-negative constraint on the model coefficients $E_{\text{abs,i}}$ was
included for physical meaning. The algorithm was applied in bootstrap with replacement of
residuals for $10^4$ runs, and convergence of the bootstrap results was checked by varying the
number of samples. The resulting estimates of mean and standard error of $E_{\text{abs}}$ for all PMF
factors are listed in Table 3.
A scatter plot of the predicted $b_{\text{abs,BrC,pred}}$ against the observed $b_{\text{abs,BrC}}$ is shown in Figure
14h. The model captured 66% of the variance in $b_{\text{abs,BrC}}$, and the PMF factor loadings can be
considered good predictors of the BrC absorption under the study conditions. Physical factors not
directly represented in this statistical model, such as the effects of mixing state, size distribution,



and so on for BrC absorption, either have low variability under the study conditions or
alternatively have co-variability also captured in the PMF factor loadings.

The highest values of $E_{abs}$ at 370 nm were associated with the HOA and LO-BBOA

factors (2.04 ± 0.14 and 1.50 ± 0.07 m$^2$ g$^{-1}$, respectively). These results support the interpretation
presented in the previous section about the association of the HOA and LO-BBOA factors with
light absorption. As a point of comparison, $E_{abs}$ of 2 to 3 m$^2$ g$^{-1}$ at 300 nm was reported for
HULIS extracts from PM$_{2.5}$ filter samples collected under biomass burning conditions during the
Amazon dry season in Rondônia, Brazil (Hoffer et al., 2006). HULIS have been recognized as
important components of BrC from biomass burning (Mukai and Ambe, 1986; Andreae and
Gelencsér, 2006; Graber and Rudich, 2006). The $E_{abs}$ value of the MO-BBOA factor was 0.82 ±
0.04 m$^2$ g$^{-1}$. The result of $E_{abs,MO-BBOA} < E_{abs,LO-BBOA}$ is consistent with an interpretation of
photochemically driven oxidation and bleaching during the atmospheric transport of biomass
burning emissions.

The $E_{abs}$ value of the IEPOX-SOA factor was 0.40 ± 0.05 m$^2$ g$^{-1}$, and the $E_{abs}$ values of

the MO-OOA and LO-OOA factors (0.01 ± 0.02 m$^2$ g$^{-1}$) were not statistically different from
zero. Laboratory studies suggest that biogenic PM does not appreciably absorb light in the near-
UV and visible range although this result may change with atmospheric exposure to ammonia
and amines, changes in particle acidity, and other factors (Nakayama et al., 2012; Liu et al.,
2013; Flores et al., 2014; Lin et al., 2014; Laskin et al., 2015). Biogenic PM is typically
characterized by carbonyls, carboxyls, and hydroxyls without substantial conjugation; this
composition does not have the low-energy electronic transitions relevant for brown-carbon light
absorption (Laskin et al., 2015). By contrast, PM produced by the photo-oxidation of aromatic
VOCs, such as toluene, $m$-xylene, naphthalene, and trimethylbenzene, tends to absorb



significantly, and the light absorption is greater for PM produced under conditions of higher $NO_x$
concentrations because of the production of nitro-aromatic compounds (Zhong and Jang, 2011;
Liu et al., 2012; Lee et al., 2014; Liu et al., 2015). This absorption, however, may decrease with
atmospheric processing as previously discussed for the case of biomass burning emissions,
which is also reflected in the negligible value of $E_{abs}$ for MO-OOA. In central Amazonia, the
organic PM is dominated by biogenic forest precursors even in the pollution plume of Manaus,
which helps to explain the negligible $E_{abs}$ value for LO-OOA. It may also be that some aromatic
PM is associated with the HOA factor, which has a high $E_{abs}$ value.

A comparison of the relative contributions of PMF factor loadings to organic $PM_1$ mass

concentration and to light absorption is presented in Figure 15 (left and right panels,
respectively). The contribution of each class of organic compounds to total absorption by organic
$PM_1$ was estimated for each point in time by multiplication of the $E_{abs}$ value and the loading of
each PMF factor during IOP2. The means and standard deviations of the resulting percentage
contributions are listed in Table 4. Biomass burning and urban emissions, as represented by the
BBOA and HOA factors, appeared to contribute 80% of $b_{abs,BrC}$ while accounting for at least
30% of the organic $PM_1$ mass concentration. The IEPOX-SOA factor was associated with the
balance of $b_{abs,BrC}$ while representing 16% of the organic $PM_1$ mass concentration. Studies with
further information on black carbon size distribution, particle mixing state, and the effect of RH
on particle absorption are warranted to refine the estimates of $E_{abs}$ for the components of organic
$PM_1$ and therefore their contributions to BrC light absorption. A similar attribution analysis as
the right panel of Figure 15 was carried out for the baseline, event, and urban clusters separately
and is discussed in the Supplementary Material (Figure S15).



**4.  Summary and Conclusions**
The influence of urban and biomass burning emissions on the otherwise natural
concentrations, composition, and optical properties of organic $PM_1$ in central Amazonia were
investigated during the dry season. Positive-matrix factorization was used to classify the organic
PM into subcomponents. The MO-OOA, LO-OOA, and IEPOX-SOA together accounted for
about 62% of the organic PM. The MO-BBOA and LO-BBOA factors together accounted for
31%, and HOA for the remaining 7%. An important conclusion is that the 8.5-fold increase in
organic $PM_1$ concentrations between the wet and dry seasons is not all due to biomass burning,
but also to a concurrent increase of biogenic secondary organic $PM_1$ of eight-fold and smaller
increases in urban $PM_1$. Reasons that possibly played a role in such increases for the dry season
are: increased BVOC emissions, increased formation potential of biogenic secondary organic
$PM_1$, reduced wet and dry deposition and PBL ventilation of $PM_1$ particles, and increased
partitioning due to larger organic $PM_1$ mass concentrations in the dry season.
The FCM clustering analysis identified the baseline, event, and urban clusters. Relative to
the baseline cluster (9.2 $\mu g\ m^{-3}$), both the event and the urban cluster had an increase of 3 $\mu g\ m^{-3}$.
For the event cluster, the increased sulfate concentrations together with only moderate increases
in $NO_y$, resulted in remarkable increases of almost 1 $\mu g\ m^{-3}$ (65%) in IEPOX-SOA factor
loadings relative to the baseline cluster. Regarding the urban cluster, increases in the factor
loadings of MO-BBOA (40 to 90%) and LO-OOA (20 to 25%) were observed in comparison to
the other two clusters. At the same time, the IEPOX-SOA contribution was either the same or
lower (by 40%) in absolute loadings, and always lower in relative contribution to organic PM
(15% of organic PM compared to 20-30% for the other clusters). These changes in the make-up
of organic PM were consistent with the changes observed for the wet season (de Sá et al., 2017;

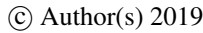



de Sá et al., 2018). They were attributed partly to (i) a shift in oxidation pathways from $HO_2$- to
NO-dominant, and partly to (ii) an accelerated oxidation cycle that increases the mass
concentration of secondary organic PM.

Optical properties of the $PM_1$ were investigated, focusing on the organic component. The

BrC absorption coefficient $b_{abs,BrC}$ had an inverse relationship with O:C ratio and a positive
relationship with the $C_xH_yO_zN_p^+$ family, indicating that BrC light in this region was associated
with less-oxidized and N-containing organic compounds. The LO-BBOA and HOA factors had
the lowest O:C ratios and highest relative contribution of $C_xH_yO_zN_p^+$ family ions, suggesting that
these factors represent BrC components. In addition, a tight relationship between $å_{abs}$ and
$\log_{10}(f_{60}/f_{44})$ was found, corroborating the importance of BBOA factors for absorption properties
of organic PM, and possibly providing a parameterization for $å_{abs}$ in the region. Further analysis
determined the $E_{abs}$ associated with each of the PMF factors. The results implied that the MO-
OOA and LO-OOA factors were associated with non-absorbing components. The MO-BBOA
($E_{abs} = 0.8$ $m^2$ $g^{-1}$), LO-BBOA (1.5 $m^2$ $g^{-1}$), and HOA (2.0 $m^2$ $g^{-1}$) factors were associated with
80% of the light absorption by organic PM in the region. The remaining absorption (<10%) was
attributed to IEPOX-SOA ($E_{abs} = 0.4$ $m^2$ $g^{-1}$).

The BrC light absorption can have direct and indirect effects on radiative forcing, which

ought to be further investigated for the Amazon region. The inclusion of BrC absorption in
models may result in a positive direct radiative forcing in regions of high BrC concentrations, in
contrast to models that assume organic PM as a purely scattering component (Ramanathan and
Carmichael, 2008; Myhre et al., 2013). Recent models have estimated the global BrC
contribution to DRF to be in the range of 0.1 to 0.25 W $m^{-2}$, corresponding to 10 to 25% of the
DRF by BC (Feng et al., 2013). In addition, BrC in cloud water can absorb light and thereby



facilitate water evaporation and cloud dispersion (Hansen et al., 1997). This effect may
compensate the cooling that aerosol particles offer by serving as seeds for cloud droplet
formation and may also provide a positive feedback as increased fire activity may provoke more
fire-prone conditions by suppressing precipitation (Nepstad et al., 1999; Bevan et al., 2009;
Gonçalves et al., 2015; Laskin et al., 2015). Another implication is that light absorption by BrC
in the ultraviolet may significantly decrease photolysis rates, thereby affecting the concentrations
of precursors and oxidants such as ozone and OH radicals in the atmosphere (Li et al., 2011;
Jiang et al., 2012; Laskin et al., 2015).

Given the importance of biomass burning and the increasing importance of urban

pollution in the Amazon forest, light absorption by atmospheric particulate matter could become
more prevalent in this region in the future. Further field, laboratory, and modeling studies are
warranted to (i) more finely map the importance of both urban and biomass burning emissions at
different locations in the Amazon region, (ii) characterize BrC components at the molecular level
for structure-absorption relationships, and (iii) quantify the effects of BrC absorption on radiative
forcing in the regional and global scales for current and future scenarios of increased human
impacts.



**Data availability**

The data sets used in this publication are available at the ARM Climate Research Facility database for the GoAmazon2014/5 experiment (https://www.arm.gov/research/campaigns/amf2014goamazon, last access: 1 August 2018).

**Author contributions**

SSdS, LVR, and STM defined the scientific questions and scope of this study. STM, JLJ, MLA, AHG, and PA designed, planned, and supervised the broader GoAmazon2014/5 field experiment. SSdS, BBP, PCJ, and DAD carried out the AMS measurements and data processing. AS collected and quality-checked the aethalometer data. LVR performed the BrC calculations based on the aethalometer data. LDY, RW, GYV, JB, SC, YJL, SS, and HMJB performed auxiliary data collection/processing and simulations. SSdS carried out the scientific analysis involving PMF and FCM. SSdS prepared the paper with contributions from all co-authors.



**Acknowledgments**

Institutional support was provided by the Central Office of the Large Scale Biosphere Atmosphere Experiment in Amazonia (LBA), the National Institute of Amazonian Research (INPA), and Amazonas State University (UEA). We acknowledge support from the Atmospheric Radiation Measurement (ARM) Climate Research Facility, a user facility of the United States Department of Energy (DOE, DE-SC0006680), Office of Science, sponsored by the Office of Biological and Environmental Research, and support from the Atmospheric System Research (ASR, DE-SC0011115, DE-SC0011105) program of that office. Additional funding was provided by the Amazonas State Research Foundation (FAPEAM 062.00568/2014 and 134/2016), the São Paulo State Research Foundation (FAPESP 2013/05014-0, 2017/17047-0 and 2013/50510-5), the USA National Science Foundation (1106400 and 1332998), and the Brazilian Scientific Mobility Program (CsF/CAPES). S. S. de Sá acknowledges support by the Faculty for the Future Fellowship of the Schlumberger Foundation. B. B. Palm acknowledges a US EPA STAR graduate fellowship (FP-91761701-0). This manuscript has not been reviewed by EPA and no endorsement should be inferred. BBP, PCJ, DAD, and JLJ were supported by DOE (BER/ASR) DE-SC0016559 and NSF AGS-1822664. Data access from the Sistema de Proteção da Amazônia (SIPAM) is gratefully acknowledged. The research was conducted under scientific license 001030/2012-4 of the Brazilian National Council for Scientific and Technological Development (CNPq).





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



**List of Tables**

**Table 1.** Characteristics of the PMF factor profiles. Listed are $f_{44}$ and $f_{60}$, corresponding to the organic signal fraction at $m/z$ 44 and $m/z$ 60, respectively, as well as the oxygen-to-carbon (O:C) and hydrogen-to-carbon (H:C) ratios. Values and uncertainties were calculated by running the PMF analysis in "bootstrap mode" (Ulbrich et al., 2009). The Pearson-$R$ correlations between the factor profiles of IOP2 and their counterparts in IOP1 are also listed (i.e., dry season compared to wet season). "N/A" means "not applicable". Elemental ratios were calibrated by the "improved-ambient" method, which has an estimated uncertainty of 12% for O:C and 4% for H:C (Canagaratna et al., 2015).

| PMF factor | $f_{44}$ | $f_{60}$ | O:C | H:C | Pearson-$R$ against IOP1 counterpart |
|---|---|---|---|---|---|
| MO-OOA | $0.24 \pm 0.01$ | $< 0.001$ | $1.20 \pm 0.10$ | $1.25 \pm 0.08$ | 1.00 |
| LO-OOA | $0.15 \pm 0.01$ | $0.001 \pm 0.001$ | $0.86 \pm 0.08$ | $1.51 \pm 0.06$ | 0.99 |
| IEPOX-SOA | $0.14 \pm 0.01$ | $< 0.001$ | $0.74 \pm 0.02$ | $1.51 \pm 0.01$ | 0.99 |
| MO-BBOA | $0.13 \pm 0.01$ | $0.011 \pm 0.003$ | $0.70 \pm 0.07$ | $1.59 \pm 0.11$ | N/A |
| LO-BBOA | $0.02 \pm 0.01$ | $0.05 \pm 0.01$ | $0.53 \pm 0.04$ | $1.79 \pm 0.06$ | N/A |
| HOA | $0.05 \pm 0.01$ | $0.001 \pm 0.001$ | $0.22 \pm 0.03$ | $1.82 \pm 0.03$ | 0.94 |





**Table 2.**  Relationship of PMF factors to organo-nitrogen characteristics. Listed for each factor are the mean loading of the time series, the percent contribution of the $C_xH_yO_zN_p^+$ family to the factor profile, the mean mass concentration of the $C_xH_yO_zN_p^+$ family (based on multiplication of columns 2 and 3), as well as the Pearson-$R$ correlation of factor loading against the mass concentration of $C_xH_yO_zN_p^+$, the mass concentration of organic nitrates, and $b_{abs,BrC}$. The $C_xH_yO_zN_p^+$ family corresponds to the sum of all ions containing at least one C atom and one N atom, as measured by the AMS. Detailed family-colored spectra showing the nitrogen-containing ions for all PMF factors are presented in Figure S6, and the most important ion fits are shown in Figure S7. The AMS method characterizes organic nitrates through the $NO^+$ and $NO_2^+$ fragments, which remain distinct from the larger fragments of the $C_xH_yO_zN_p^+$ family (Section S1 and discussion therein).

| PMF factor | Mean factor loading ($\mu g\ m^{-3}$) | Nitrogen characteristics of factor profile | | | Pearson $R$ of factor loading | | |
| --- | --- | --- | --- | --- | --- | --- | --- |
| | | $C_xH_yO_zN_p^+$ family contribution (%) | Mass concentration of the $C_xH_yO_zN_p^+$ family ($\mu g\ m^{-3}$) | | Against the mass concentration of $C_xH_yO_zN_p^+$ family | Against the mass concentration of organic nitrates | Against $b_{abs,BrC}$ |
| MO-OOA | 1.6 | 5.7 | 0.09 | | 0.33 | 0.38 | 0.17 |
| LO-OOA | 2.2 | 3.7 | 0.08 | | 0.10 | 0.15 | -0.19 |
| IEPOX-SOA | 1.2 | 6.6 | 0.08 | | 0.39 | 0.40 | 0.17 |
| MO-BBOA | 1.5 | 2.9 | 0.04 | | 0.65 | 0.24 | 0.53 |



| LO-BBOA | 1.0 | 10.4 | 0.11 | 0.89 | 0.13 | 0.69 |
|---|---|---|---|---|---|---|
| HOA | 0.6 | 9.0 | 0.05 | 0.82 | 0.20 | 0.68 |

**Table 3.** Results of the constrained linear least squares regression analysis for the brown-carbon absorption coefficient (Equation 5). (a) Mass absorption efficiency $E_{abs}$ associated with each PMF factor. (b) Model intercept. The mean, standard error (SE), and 95% confidence interval (CI) are listed in each panel. Unit of Mm$^{-1}$ represents $10^{-6}$ m$^{-1}$. The coefficient of determination $R^2$ between predicted $b_{abs,BrC,pred}$ and observed $b_{abs,BrC}$ was 0.66. The symbol "*" indicates that the estimated value was statistically not higher than zero.

| (a) | $E_{abs}$ (m$^2$ g$^{-1}$) | | |
|---|---|---|---|
| PMF factors | Mean | SE | CI |
| MO-OOA | 0.01* | 0.02 | [0.00, 0.08] |
| LO-OOA | 0.01* | 0.02 | [0.00, 0.08] |
| IEPOX-SOA | 0.40 | 0.05 | [0.31, 0.50] |
| MO-BBOA | 0.82 | 0.04 | [0.75, 0.90] |
| LO-BBOA | 1.50 | 0.07 | [1.37, 1.63] |
| HOA | 2.04 | 0.14 | [1.76, 2.31] |

| (b) | $b_{abs}$ (Mm$^{-1}$) | | |
|---|---|---|---|
| Model intercept | Mean | SE | CI |
| $B$ | 0.13* | 0.10 | [0.00, 0.33] |



**Table 4.**   Contribution of PM$_1$ components as represented by the PMF factors to organic mass concentrations and BrC light absorption. The contribution of the model intercept to BrC light absorption is also included. Values listed are resulting means and standard deviations of the contributions calculated throughout IOP2. Small differences between the values in column 2 and the values represented in the inset of Figure 5a are due to differences in data coverage by the aethalometer and AMS.

| PMF factor | Contribution to organic mass concentration (%) | Contribution to BrC light absorption (%) |
|---|---|---|
| MO-OOA | 21.1 ± 10.0 | 0.5 ± 0.4 |
| LO-OOA | 30.9 ± 11.4 | 0.8 ± 0.5 |
| IEPOX-SOA | 16.3 ± 9.8 | 15.7 ± 11.2 |
| MO-BBOA | 16.7 ± 12.0 | 28.9 ± 18.0 |
| LO-BBOA | 9.5 ± 7.5 | 27.8 ±14.3 |
| HOA | 5.5 ± 3.9 | 21.7 ± 10.5 |
| Model intercept | N/A | 4.6 ± 2.6 |





**List of Figures**

**Figure 1.** $PM_1$ composition during the dry season from August 15 to October 15, 2014, representing the second Intensive Operating Period (IOP2) of the GoAmazon2014/5 experiment. Results are shown for measurements at T3 in comparison to other sites. (a) $PM_1$ mass concentrations of non-refractory AMS organic, sulfate, ammonium, nitrate, and chloride. Mass concentrations of SP2 refractory black carbon (rBC) are also plotted. rBC refers to the carbon content of graphite-like components that are strongly light-absorbing (Pöschl, 2003). (b) (Top) Summed mass concentrations and (bottom) segregated mass fractions of the non-refractory species at the T0a, T2, and T3 sites. The inset of panel a shows the locations of the relevant research sites for this study. A larger map is shown in Figure 3. T0a is the Amazonian Tall Tower Observatory (Andreae et al., 2015). T2 is a site 8 km downwind of Manaus, just across the Black River ("Rio Negro") (Cirino et al., 2018). Measurements at T0a and T2 were made by an ACSM. Concentrations in both panels were adjusted to standard temperature (273.15 K) and pressure ($10^5$ Pa) (STP).

**Figure 2**. Time series of (a) organic and (b) sulfate mass concentrations at the T0a, T2, and T3 sites. Concentrations were adjusted to standard temperature (273.15 K) and pressure ($10^5$ Pa).

**Figure 3.** Fire locations in the upwind region of the T3 site for each week of IOP2. Transport times from the fires to the T3 site represent up to 15 h at the scale of this figure and typical wind speeds. The plotted data was obtained from the fire database of the Brazilian National Institute of Spatial Research (INPE, 2018). Underlying image: Google Maps.



**Figure 4.** Diel trends of (top) organic and (bottom) sulfate mass concentrations at the T0a, T2, and T3 sites. Lines represent means, solid markers show medians, and boxes span interquartile ranges. Local time is UTC minus 4 h. Concentrations were adjusted to standard temperature (273.15 K) and pressure ($10^5$ Pa).

**Figure 5.** PMF analysis of the time series of AMS organic mass spectra collected at the T3 site. (a) Mass spectral profile of each factor represented at unit mass resolution. The inset shows the mean fractional loading of each factor. The factor profiles are shown in more detail, colored by ion families, in Figure S5. (b) Diel trends for the loadings of each PMF factor. Local time is UTC minus 4 h. Lines represent means, solid markers show medians, and boxes span interquartile ranges. (c) Time series of the factor loadings.

**Figure 6.** Pearson-$R$ correlations between the loading of each PMF factor and concentrations of selected measurements at the T3 site. Abbreviations include tricarballylic acid (TCA), methyl-butyl-tricarboxylic acid (MBTCA), methyl vinyl ketone (MVK), methacrolein (MACR), isoprene hydroxyhydroperoxides (ISOPOOH), and refractory black carbon (rBC). SV-TAG measurements refer to particle-phase concentrations, except for sesquiterpenes which refer to total concentrations and mostly occurred in the gas phase. The $C_8$ and $C_9$ aromatics include the xylene and trimethylbenzene isomers, respectively. The $C_{20}$, $C_{22}$, and $C_{24}$ acids include eicosanoic, docosanoic, and tetracosanoic acids, respectively.

**Figure 7.** Analysis of the organic $PM_1$ sampled at the T3 site in relation to biomass burning. (a) Scatter plot of the AMS signal fraction at $m/z$ 44 ($f_{44}$) against that at $m/z$ 60 ($f_{60}$). Red circles represent measurements during the dry season (IOP2), and blue squares



represent measurements at the same site during the wet season (IOP1) (de Sá et al., 2018). Diamonds represent the MO-BBOA and LO-BBOA factors of IOP2. The dashed line represents a reference for negligible influence by biomass burning based on several field studies (Cubison et al., 2011). (b) Diel trends of the fractional loadings of the MO-BBOA and LO-BBOA factors relative to their sum BBOA$_T$. Local time is UTC minus 4 h.

**Figure 8.** Results of the cluster analysis by Fuzzy c-means (FCM) for afternoon periods (12:00 to 16:00) presented by several case studies. The shown case studies represent 30% of the FCM results. (a) Degree of membership in each of the three clusters. The sum of degrees of membership across all clusters is unity. (b) Pollution indicators: concentrations of NO$_y$, O$_3$, CO, refractory black carbon (rBC), and particle number count are plotted. (c) PM$_1$ mass concentrations for organic, sulfate, nitrate, and ammonium species. (d) Fractional contribution of each factor to the PM$_1$ organic mass concentration.

**Figure 9.** Air-mass backtrajectories associated with the three clusters of the FCM analysis. Trajectories were calculated using HYSPLIT4 in steps of 12 min for 10 h (Draxler and Hess, 1998). Twenty trajectories are plotted for each cluster, corresponding to the times of highest degree of membership to that cluster.

**Figure 10.** PM$_1$ characterization represented by the centroids of the FCM clusters. (a) Mass concentrations of AMS species. (b) PMF factor loadings. Results are for afternoon time periods. Table S1 lists the values presented in this figure.

**Figure 11.** Diel trends of PM$_1$ optical properties. (a) Total absorption coefficient $b_{abs}$ (370 nm). (b) Absorption coefficient $b_{abs,BrC}$ of brown carbon (370 nm). (c) Fractional



contribution of $b_{\text{abs,BrC}}$ to $b_{\text{abs}}$. (d) Absorption Ångstrom exponent $\mathring{a}_{\text{abs}}$ from 370 to 430 nm. Local time is UTC minus 4 h.

**Figure 12.** Relationships between the brown-carbon absorption coefficient and the organic PM$_1$ composition. Scatter plots of $b_{\text{abs,BrC}}$ against (a) the oxygen-to-carbon ratio (O:C) and (b) the mass concentration of the nitrogen-containing $C_xH_yO_zN_p^+$ family. For the $C_xH_yO_zN_p^+$ family, all ions contain at least one C atom and one N atom, meaning $x > 0$, $y \geq 0$, $z \geq 0$, and $p > 0$. Boxes indicate interquartile ranges, and horizontal lines within the boxes indicate medians. In complement, Figure S14 shows the relationships between the brown-carbon absorption coefficient and the fractional contributions of the $C_xH_yO_z^+$ and $C_xH_yO_zN_p^+$ families to organic PM$_1$.

**Figure 13.** Relationships between the absorption Ångstrom exponent and indicators of biomass burning. Scatter plots of $\mathring{a}_{\text{abs}}$ against (a) $\log_{10}(f_{60}/f_{44})$ of the AMS analysis ($R = 0.87$), and (b) the ratio of the BBOA$_T$ loading to the organic PM$_1$ mass concentration ($R = 0.75$). BBOA$_T$ loading is the sum of the MO-BBOA and LO-BBOA factor loadings. The $\mathring{a}_{\text{abs}}$ value corresponds to 370 to 430 nm. In panel a, the slope and intercept are $3.2 \pm 0.1$ and $6.8 \pm 0.1$, respectively. In panel b, they are $5.2 \pm 0.1$ and $1.1 \pm 0.1$.

**Figure 14.** Scatter plots against $b_{\text{abs,BrC}}$ of (a-f) PMF factor loadings, (g) organic PM$_1$ mass concentration, and (h) $b_{\text{abs,BrC,pred}}$ values predicted by a multivariate linear regression model using PMF factor loadings as parameters as described by Equation 5.

**Figure 15.** Comparative relationship of the relative contributions of PMF factor loadings to (left) organic PM$_1$ mass concentration and (right) organic PM$_1$ light absorption. Results represent means for the full datasets of IOP2. The means and standard





deviations are listed in Table 4. Results are for 370 nm. "Other" refers to the model

intercept $B$ (Equation 5).





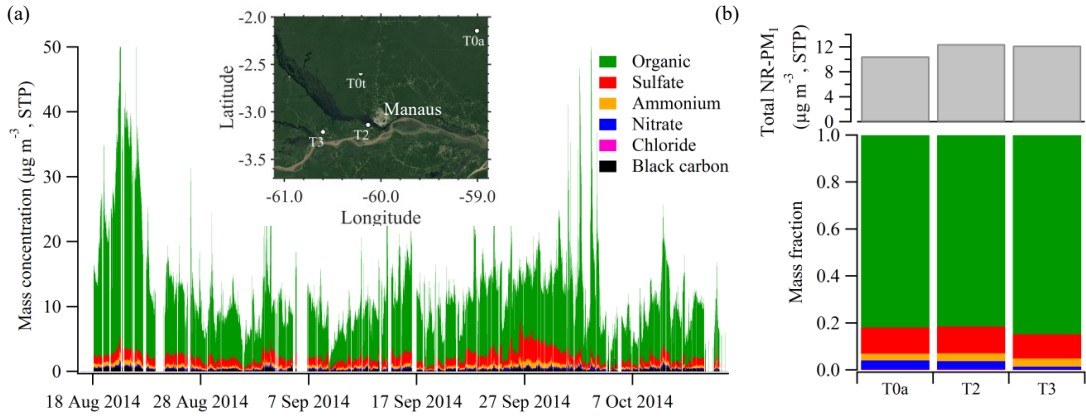





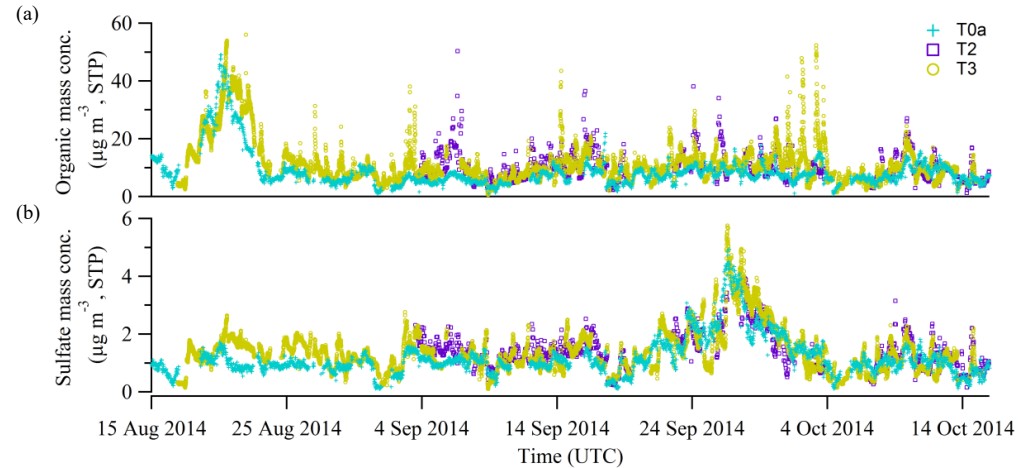

*Figure 2*





*Figure 3*



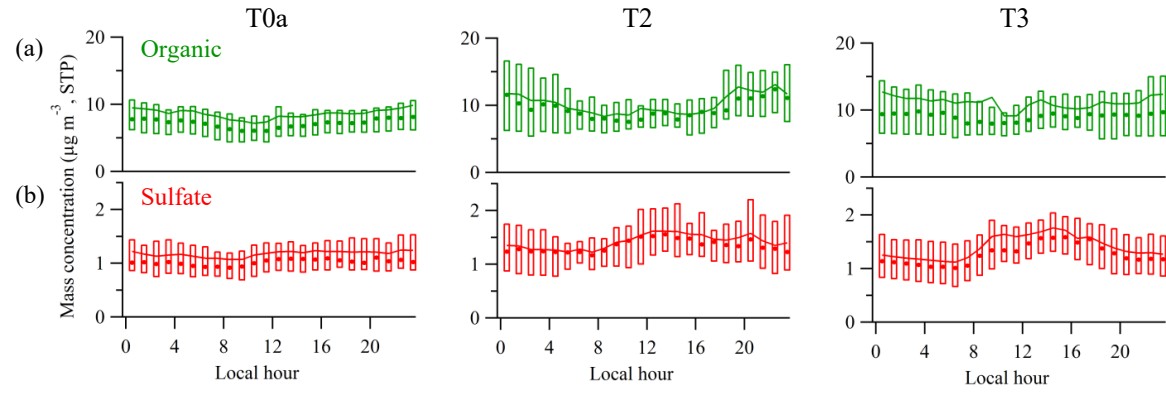

*Figure 4*





*Figure 5*



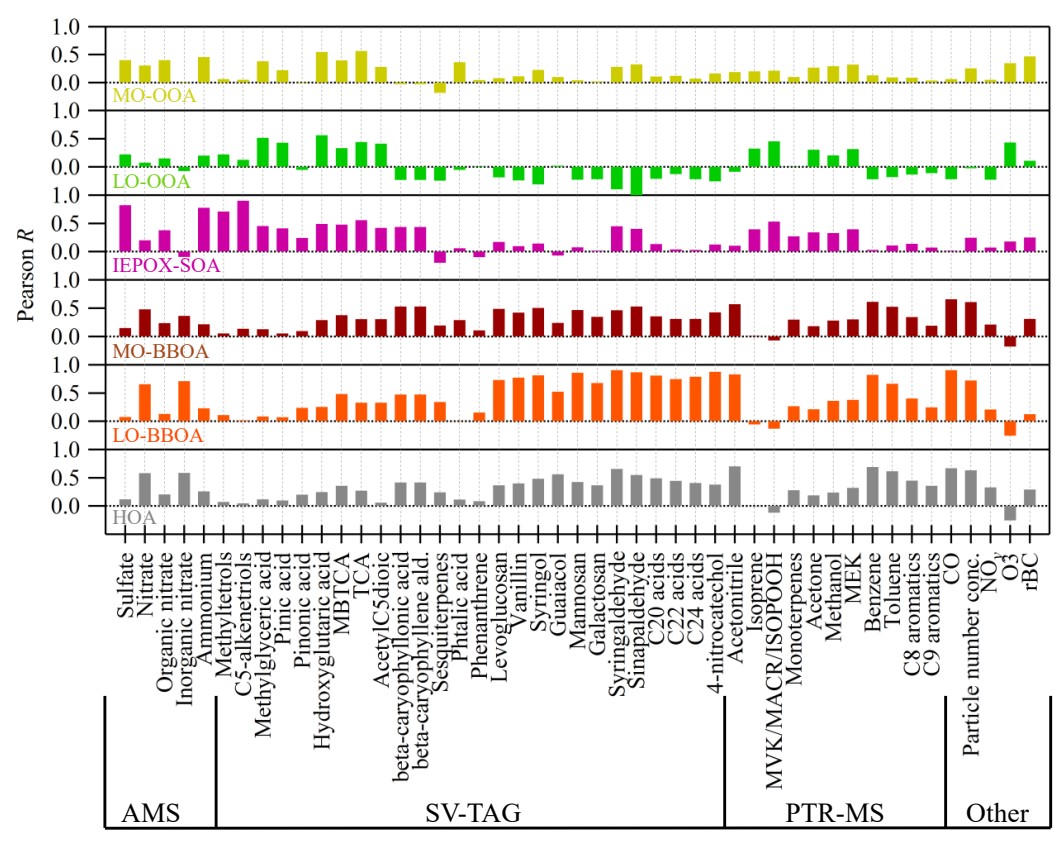

*Figure 6*



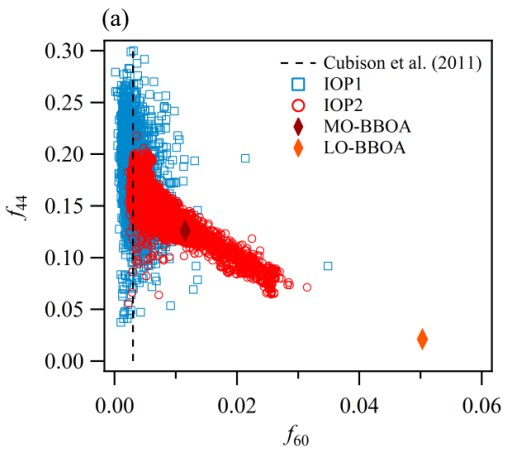
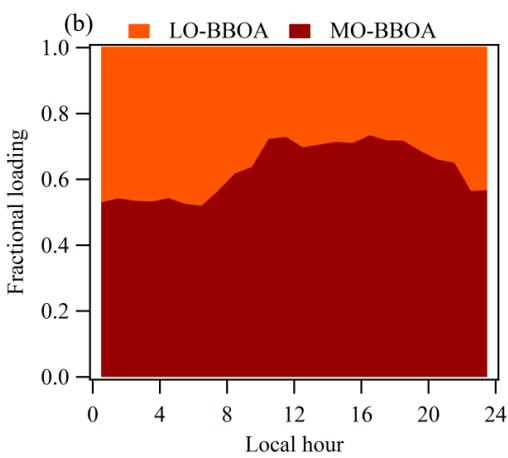

*Figure 7*





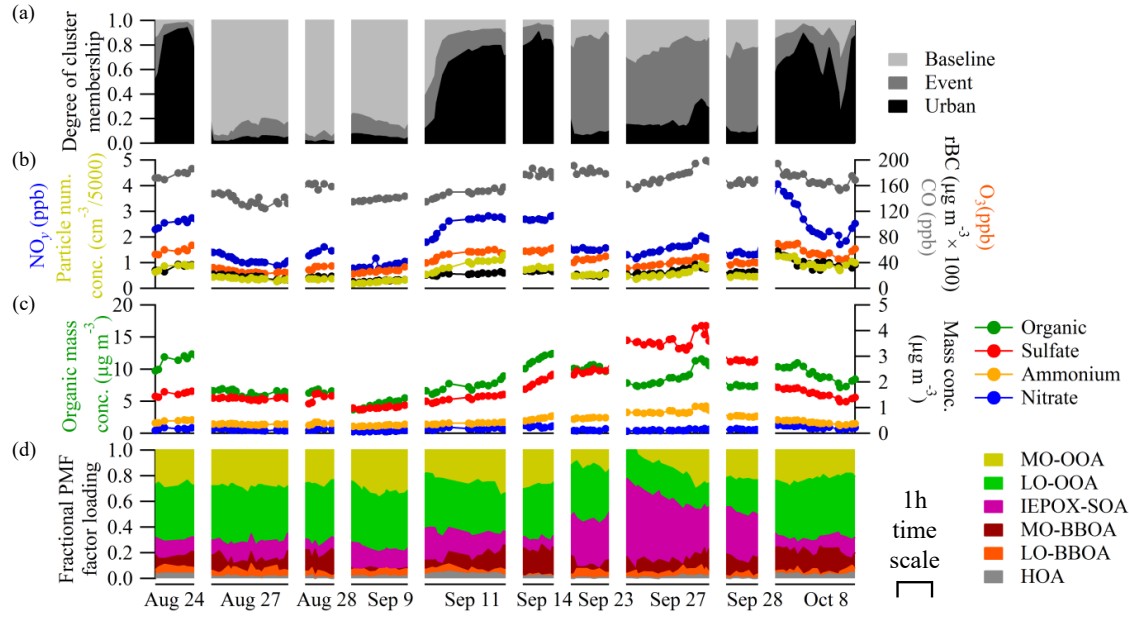

*Figure 8*



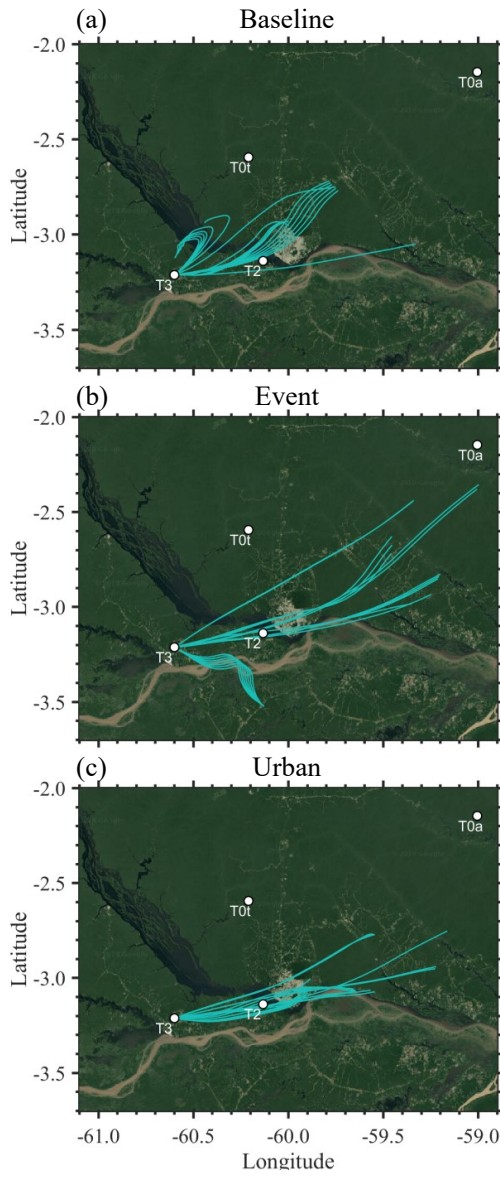

*Figure 9*





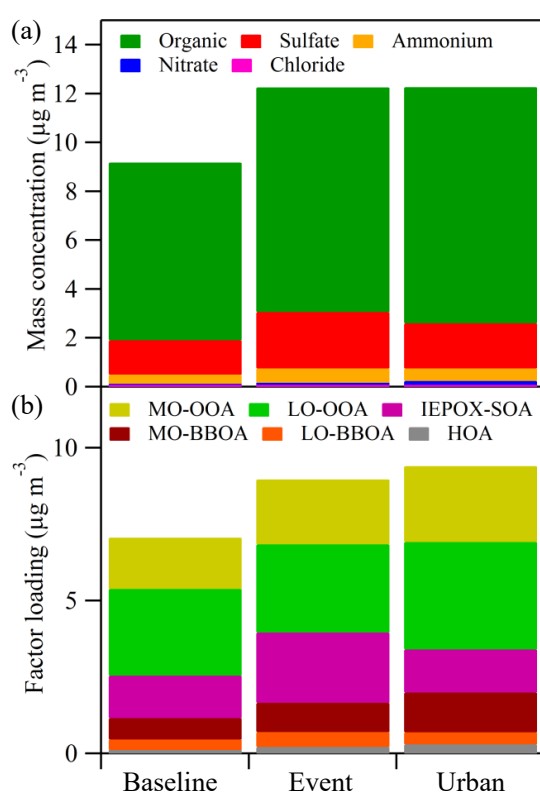

*Figure 10*



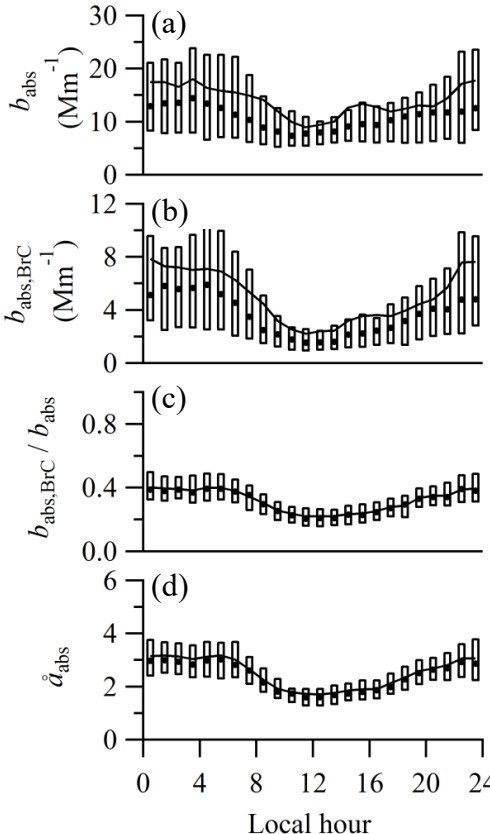

Figure 11





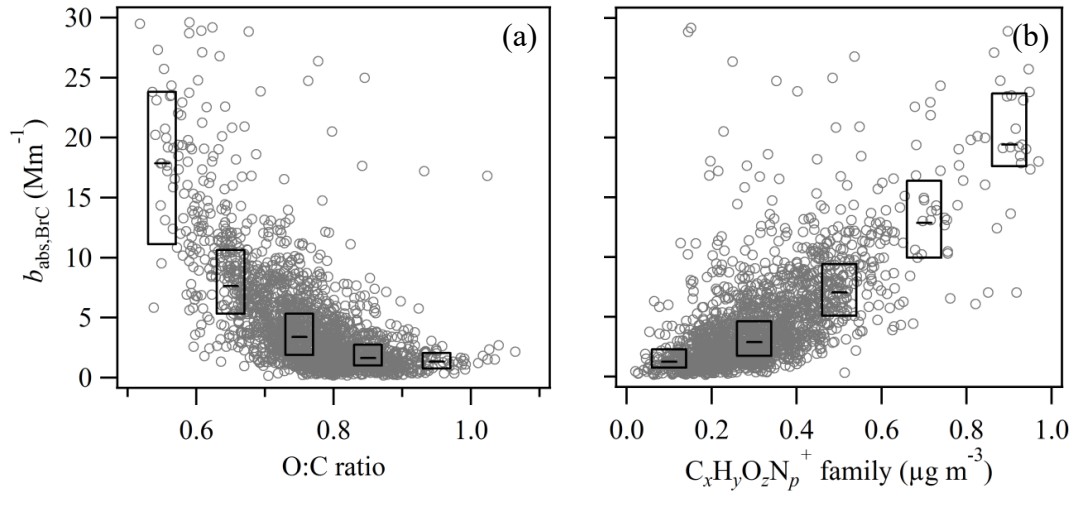

*Figure 12*



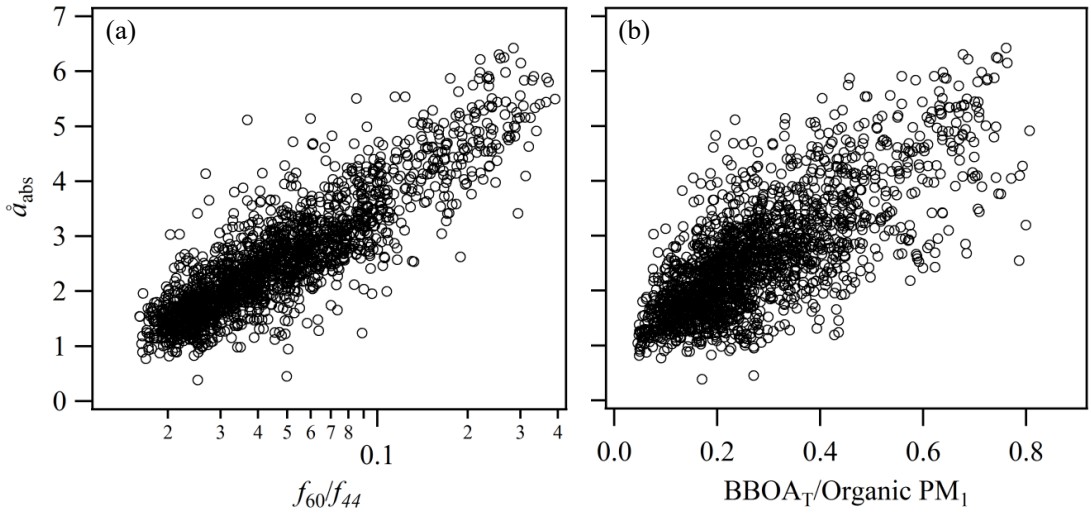

*Figure 13*





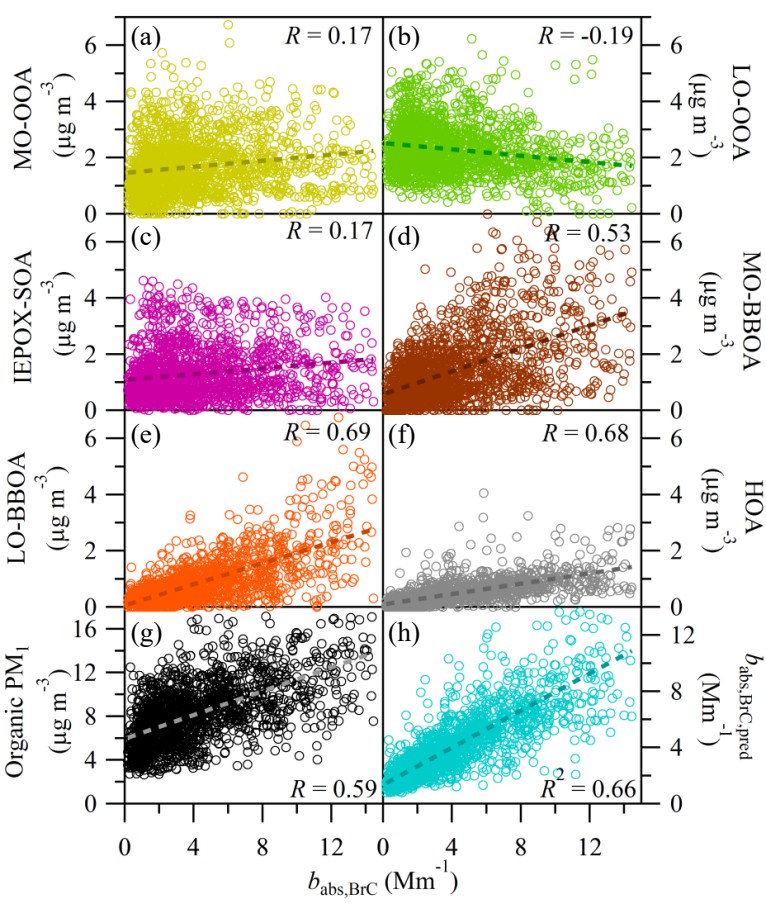

*Figure 14*





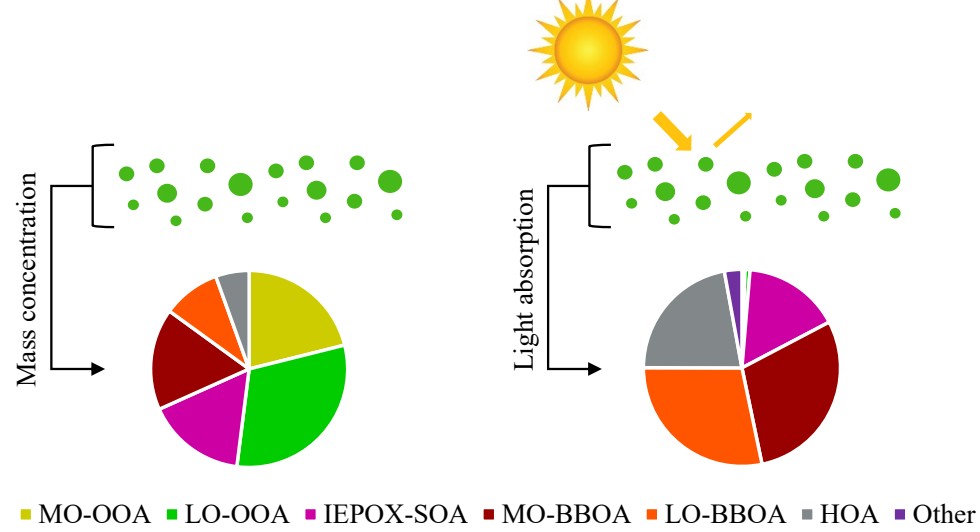

*Figure 15*