# Peer review of "Contributions of biomass-burning, urban, and biogenic emissions to the concentrations and light-absorbing properties of particulate matter in central Amazonia during the dry season"

_Atmospheric Chemistry and Physics, 2018_

## Short Comment (SC1) · 15 Jan 2019

I would like to offer a few comments based on a quick reading.

In Section 3.2.2 the authors estimated the mass absorption efficiency (MAE) of different PMF ACSM factors using multivariate linear regression (MLR). Using PMF factors in the MLR, rather than individual chemical species, has the great advantage of providing the MAE of atmospheric particles taking into consideration their mixing state in the

atmosphere. However, as reported by the authors, the number of papers presenting MAE (or mass scattering efficiency; MSE) of pollutant sources is rather scarce. Here, I would like to suggest the authors to cite another recent paper (Ealo et al., ACP, 2018) where both the MAE and MSE of different PMF sources were reported. The chemical speciated data used in Ealo et al. (2018) were obtained from chemical analysis of 24h filters. In Ealo et al. (2018) the highest MAE was calculated for the Traffic source (around 1.7 m2/g at 637 nm).

Moreover, Ealo et al. (2018) also reported the correlation between measured and modelled aerosol particle scattering (R2 = 0.85) and absorption (R2 = 0.76).

Ealo, M., Alastuey, A., Pérez, N., Ripoll, A., Querol, X., and Pandolfi, M.: Impact of aerosol particle sources on optical properties in urban, regional and remote areas in the north-western Mediterranean, Atmos. Chem. Phys., 18, 1149-1169, https://doi.org/10.5194/acp-18-1149-2018, 2018.
* * *

---

## Referee Comment (RC1) · Anonymous Referee #1 · 5 Feb 2019

1. Line 155. As reported by (Zotter et al 2017) at 370 nm there maybe a non-negligible light-absorption contribution from SOA compounds, I would use the wavelength at 430 nm. 2. Line 166-170. I would use some references to support the assumptions made. 3. Line 528-537. When you describe Figure 11 you don't comment on the Angström exponent 4. Figure 4 and Figure 11: I would write the values of the interquartile ranges: 25, 75 or 10, 90? 5. Figure 11: I would write the wavelengths also on the graph to be clear, in particular on the Angstrom exponent 6. Figure 13: I would draw the correlation

curve and write the correlation coefficient as in Figure 14.

---

## Referee Comment (RC2) · Anonymous Referee #2 · 19 Feb 2019

De Sá et al. have studied concentrations and light absorption properties of PM during the dry season in central Amazonia, as part of the GoAmazon2014/15 campaign. They present a wealth of data and analyze it in a comprehensive and detailed fashion to derive some interesting insights on anthropogenic impacts on PM and OA concentrations as well as light absorption properties over Amazonia. The paper is well written, the Figures are numerous, but clear and mostly justified (see comment below) and the conclusions are well-based on the measured data. The abstract could be improved

(see comment below) and I have a few more comments listed below. I recommend publication of the paper after these have been addressed.

Comments:

- The abstract is rather descriptive and would benefit from more detailed quantitative results, e.g. on the measured concentrations, PMF factor contributions to OA, and contribution to BrC. Quantitative results would also be important to better understand some of the core findings mentioned in the abstract, e.g. on the BrC bleaching (L13), the relevance of sources other than BB (L17-19), and the suggested different oxidation pathways in the different clusters (L29-31). In turn, the parts that just describe what has been done could be condensed (e.g. L5-10, L13-15, L22-26, . . .)

- L142: Calculating a new trajectory every 12 min seems like quite a high frequency to me. Do they change at all within such short time? Also, here it says that 48 h back trajectories were calculated while the caption of Figure 9 says 10 h.

- L171-174: I cannot follow here. Are all the parameter subscripts correct? If so, please be more specific.

- L209: "highly correlated" is not very precise. Please provide Persons r. Also, if OA and sulfate are really "highly correlated", does it imply common sources?

- L211-212: I have difficulties resolving timescales of less than a day in Figure 2. Please also give r.

- L409: It is only at the very end of the discussion on possible drivers of lower concentrations during the wet season, that wet deposition is cautiously mentioned. To me this seems to be the most obvious and maybe also most relevant factor, as it efficiently removes both particles and precursor gases. Are there studies quantifying the effect of wet deposition in the area?

- L729-743: These paragraphs in the Summary seem to add new aspects to the discussion of BrC that were not addressed before. I think they would fit better into the

Results section, which would also help to shorten the quite long Summary section.

- Table 3: Statistical significance is mentioned in the caption, please include the significance level used (alpha = 0.05?) and ideally also the p-values of the model coefficients (i.e. Eabs).

- Figure 12: Please indicate the binwidth used for the boxes.

- In order to somewhat reduce the quite high number of Figures, the authors could consider to move Figure 3 to the SI, as it does not present any results, and to remove Figure 15, which just seems to be a visual repetition of Table 4.

- SI: As the SI will not get further typesetting I recommend giving captions together with the Figures, instead of listing them separately.

---

## Author Comment (AC1) · 14 Apr 2019

**Response to reviews**

Reviewer comments are in **bold**. Author responses are in plain text. Excerpts from the manuscript are in *italics*. Modifications to the manuscript are in *blue italics*. Page and line numbers in the responses correspond to those in the ACPD paper.

**Review #1**

**1. Line 155. As reported by (Zotter et al 2017) at 370 nm there maybe a non-negligible light-absorption contribution from SOA compounds, I would use the wavelength at 430 nm.**

In our manuscript, the assessment of brown carbon absorption aimed exactly at quantifying the contribution of all organic material, including SOA, to light absorption. Therefore, the choice of the lower bound of wavelength measured (370 nm) followed as the most appropriate.

Previous studies in Amazonia also used the 370 nm wavelength to estimate the brown carbon contribution to particle light absorption (e.g., Wang et al., 2016; Saturno et al., 2018). Zotter et al. (2017) had a different goal compared to our study, aiming to apportion the contribution of traffic and wood burning aerosols to light absorption, excluding the possible contribution of SOA to light absorption.

**2. Line 166-170. I would use some references to support the assumptions made.**

We appreciate this idea, and have added the following references in the text:

Line 166:

*"... (1) babs,BC(700) 166 = babs(700) and babs,BC(880) = babs(880) assuming that babs,BrC = 0 at red wavelengths (e.g., Andreae and Gelencsér, 2006; Wang et al., 2016), (2) åabs,BC(700,880) was calculated from Equation 2 using babs,BC(700) and babs,BC(880), (3) babs,BC(370) was calculated from Equation 2, using babs,BC(880) and åabs,BC(370,880) = åabs,BC(700,880) under the assumption that åabs,BC was independent of wavelength (e.g., Andreae and Gelencsér, 2006; Moosmüller et al., 2009), and finally (4) babs,BrC(370) was obtained by Equation 1 using babs,BC(370) and babs(370).*

**3. Line 528-537. When you describe Figure 11 you don't comment on the Angström exponent**

We thank the reviewer for catching this fault in completeness. We have adjusted the text as follows:

Line 534:

*"… from 0.2 in the day to 0.4 at night. The absorption Angström exponent åabs followed a similar diel trend, on average ranging from 2 during the day to 3 during the night (Figure 11d). Compared to the diel trends….*

**4. Figure 4 and Figure 11: I would write the values of the interquartile ranges:**

**25, 75 or 10, 90?**

Our understanding is that an interquartile range is, by definition, the middle 50%, i.e. from 25% to 75%. Therefore, we have opted to keep the captions short.

**5. Figure 11: I would write the wavelengths also on the graph to be clear, in particular on the Angstrom exponent**

We appreciate this suggestion. However, in the interest of keeping the figure clean, the wavelengths are not added to the graph. We believe that this clarity is provided in the figure caption, where all wavelengths are explicitly mentioned. Please also note that all the variables plotted were explicitly defined in the Section 2.2 (Brown carbon light absorption), lines 173-174.

**6. Figure 13: I would draw the correlation curve and write the correlation coefficient as in Figure 14.**

To address this great suggestion by the reviewer, we have modified the figure to include the trend lines. The Pearson R values and the equation coefficients are provided in the figure caption.
* * *
**Review #2**

**De Sá et al. have studied concentrations and light absorption properties of PM during the dry season in central Amazonia, as part of the GoAmazon2014/15 campaign. They present a wealth of data and analyze it in a comprehensive and detailed fashion to derive some interesting insights on anthropogenic impacts on PM and OA concentrations as well as light absorption properties over Amazonia. The paper is well written, the Figures are numerous, but clear and mostly justified (see comment below) and the conclusions are well-based on the measured data. The abstract could be improved (see comment below) and I have a few more comments listed below. I recommend publication of the paper after these have been addressed.**

We thank the reviewer for reading our manuscript and providing this valuable feedback.

**Comments:**
**1. The abstract is rather descriptive and would benefit from more detailed quantitative results, e.g. on the measured concentrations, PMF factor**

**contributions to OA, and contribution to BrC. Quantitative results would also be important to better understand some of the core findings mentioned in the abstract, e.g. on the BrC bleaching (L13), the relevance of sources other than BB (L17-19), and the suggested different oxidation pathways in the different clusters (L29-31). In turn, the parts that just describe what has been done could be condensed (e.g. L5-10, L13-15, L22-26, …)**

This feedback is highly appreciated. We have adjusted the abstract as follows:

*Urbanization and deforestation have important impacts on atmospheric particulate matter (PM) over Amazonia. This study presents observations and analysis of submicron PM$_1$ concentration, composition, and optical properties in central Amazonia during the dry season, focusing on the anthropogenic impacts.  The primary study site was located 70 km  downwind of Manaus, a city of over two million people in Brazil, as part of the GoAmazon2014/5 experiment A high-resolution time-of-flight aerosol mass spectrometer (AMS) provided data on PM$_1$ composition, and aethalometer measurements were used to derive the absorption coefficient b$_{abs,BrC}$ of brown carbon (BrC) at 370 nm. Non-refractory PM$_1$ mass concentrations averaged 12.2 µg m$^{-3}$ at the primary study site, dominated by organics (83%) and sulfate (11%). A decrease in b$_{abs,BrC}$ was observed as the mass concentration of nitrogen-containing organic compounds decreased and the organic PM$_1$ O:C ratio increased, suggesting atmospheric bleaching of the BrC components.  The organic PM$_1$ was separated into six different classes by positive-matrix factorization (PMF), and the mass absorption efficiency E$_{abs}$ associated with each factor was estimated through multivariate linear regression of b$_{abs,BrC}$ on the factor loadings.  The largest E$_{abs}$ values were associated with urban (2.04 ± 0.14 m$^2$ g$^{-1}$) and biomass burning (0.82 ± 0.04 m$^2$ g$^{-1}$ to 1.50 ± 0.07 m$^2$ g$^{-1}$) sources. Together, these sources  contributed at least 80% of b$_{abs,BrC}$ while accounting for 30 to 40 % of the organic PM$_1$ mass concentration. In addition, a comparison of organic PM$_1$ composition between wet and dry seasons revealed that only part  of the nine-fold increase in mass concentration between the seasons can be attributed  to biomass burning. Biomass-burning factor loadings increased by thirty-fold, elevating its relative contribution to organic PM$_1$ from about 10% in the wet season to 30% in the dry season. However, most of the PM$_1$ mass (>60%) in both seasons was accounted for by biogenic secondary organic sources, which in turn showed an eight-fold seasonal increase in factor loadings.  A combination of decreased wet deposition and increased emissions and oxidant concentrations, as*

*well as a positive feedback on larger mass concentrations are thought to play a role in the observed increases. Furthermore, Fuzzy c-means clustering identified three clusters, namely "baseline", "event", and "urban" to represent different pollution influences during the dry season. , including "baseline" (dry season background, which includes biomass burning), "event"(increased influence of biomass burning and long-range transport of African volcanic emissions), and "urban" (Manaus influence on top of the background). The baseline cluster, representing the dry season background, was associated with a mean mass concentration of $9 \pm 3 \ \mu g \ m^{-3}$. This concentration increased on average by $3 \ \mu g \ m^{-3}$ for both the urban and the event clusters. The event cluster, representing an increased influence of biomass burning and long-range transport of African volcanic emissions, was characterized by remarkably high sulfate concentrations. The urban cluster, representing the influence of Manaus emissions on top of the baseline, was characterized by an organic $PM_1$ composition that differed from the other two clusters. The differences discussed Differences in the organic $PM_1$ composition for the urban cluster compared to the other two clusters suggested a shift in oxidation pathways as well as an accelerated oxidation cycle due to urban emissions, in agreement with findings for the wet season.*

**2. L142: Calculating a new trajectory every 12 min seems like quite a high frequency to me. Do they change at all within such short time? Also, here it says that 48 h back trajectories were calculated while the caption of Figure 9 says 10 h.**

We thank the reviewer for bringing up these points. We have modified the text to clarify them as follows:

Line 141:
*Air-mass backtrajectories were estimated using HYSPLIT4 (Draxler and Hess, 1998). Data sets of the S-band radar of the System for Amazon Protection (SIPAM) in Manaus (Machado et al., 2014) provided precipitation data, which allowed to filter out trajectories that intercepted precipitation. The HYSPLIT4 sSimulations started at 100 m above T3 and were calculated up to two days back in time for every 12 min to match with the radar data up to two days back in time. Input meteorological data to the simulations were obtained on a grid of 0.5° × 0.5° were obtained from the Global Data Assimilation System (GDAS). Precipitation along the trajectories was based on data sets of the S-band radar of the System for Amazon Protection (SIPAM) in Manaus (Machado et al., 2014). Additional information on the backtrajectory calculations and on the radar were described in de Sá et al. (2018).*

Figure 9 caption:
*Trajectories were calculated using HYSPLIT4 in steps of 12 min and are shown for 10 h (Draxler and Hess, 1998).*

**3. L171-174: I cannot follow here. Are all the parameter subscripts correct? If so, please be more specific.**

We really appreciate that the reviewer caught this typo on the subscripts. We have corrected the issue as follows:

Line 172:
*Based on $b_{abs,\cancel{BC}}(370)$ and $b_{abs,\cancel{BC}}(430)$, $å_{abs}(370,430)$ was estimated.*

**4. L209: "highly correlated" is not very precise. Please provide Persons r. Also, if OA and sulfate are really "highly correlated", does it imply common sources?**

Pearson R values were added to the text as shown below. In addition, we did not mean to say that organic and sulfate are highly correlated to each other. Rather, we meant that each of those species have their concentrations well correlated across sites. The text was adjusted to eliminate this ambiguity:

Line 209:
*The time series of organic  mass concentrations across the three sites were well  correlated  over the two months when considering the timescale of a day (Figure 2a; 0.55 < R < 0.85). Similar behavior was observed for sulfate mass concentrations (Figure 2b; 0.86 < R < 0.93). The T0 and T3…*

**5. L211-212: I have difficulties resolving timescales of less than a day in Figure 2. Please also give r.**

The R values were added to the text as follows:
Line 211:
*The  data also shows that for timescales of an hour  the sites were less correlated (0.70 <R < 0.80 for sulfate, and 0.38 < R < 0.75 for organic mass concentrations).*

**6. L409: It is only at the very end of the discussion on possible drivers of lower concentrations during the wet season, that wet deposition is cautiously mentioned. To me this seems to be the most obvious and maybe also most relevant factor, as it efficiently removes both particles and precursor gases. Are there studies quantifying the effect of wet deposition in the area?**

The reviewer brings up an excellent point about the way we presented our arguments. We reorganized the text as follows below, bringing the sentences on wet deposition to the beginning of the paragraph, in order to reflect its importance in the discussion. We also added a few more references that show different deposition patterns during the wet and dry seasons.

While it is known that wet deposition is important in regulating atmospheric concentrations in Amazonia, we are not aware of any quantitative field or modelling study on the effect of wet deposition on said concentrations. The challenge arises from the confounding nature of all the processes and sources that change simultaneously between the wet and dry seasons, making it difficult to apportion the reasons for differences in aerosol concentrations.

Line 395:

*Therefore, reasons other than increased biomass burning in the dry season must have played a role in increasing organic PM1 concentrations. Importantly, the mass concentrations of sulfate and ammonium also increased by six-fold between seasons (Figure S10), indicating that atmospheric physical processes governing particle mass concentrations possibly played an important role. In this context, reduced wet deposition due to reduced convection in the dry season may be  one important contributor to the organic PM1 increases (Machado et al., 2014; Nunes et al., 2016; Chakraborty et al., 2018;Trebs et al., 2006; Pauliquevis et al., 2012). Another  aspect is that BVOC emissions are typically higher…*

**7. L729-743: These paragraphs in the Summary seem to add new aspects to the discussion of BrC that were not addressed before. I think they would fit better into the Results section, which would also help to shorten the quite long Summary section.**

We really like this suggestion and have moved the referred paragraph (lines 729-743) to the end of Section 3.2.2:

Line 689:
*... and is discussed in the Supplementary Material (Figure S15).*
*The BrC light absorption can have direct and indirect effects on radiative forcing...*

**8. Table 3: Statistical significance is mentioned in the caption, please include the significance level used (alpha = 0.05?) and ideally also the p-values of the model coefficients (i.e. Eabs).**

We have adjusted the caption to address this suggestion. Note that the confidence intervals provided for $E_{abs}$ values were stated as 95%. It thus follows that the significance level is 5%, and we have now explicitly added that to the caption. We have also reiterated from the text that the values listed were found by running the constrained least squares regression in bootstrap:

*Table 3. Results of the constrained linear least squares regression analysis for the brown-carbon absorption coefficient (Equation 5). (a) Mass absorption efficiency $E_{abs}$ associated with each PMF factor. (b) Model intercept. The mean, standard error (SE), and 95% confidence interval (CI) are listed in each panel. They were obtained through bootstrap of the regression analysis considering different samples (i.e., sets of points in time) for $10^4$ runs. Unit of $Mm^{-1}$ represents $10^{-6}$ $m^{-1}$. The coefficient of determination $R^2$ between predicted $b_{abs,BrC,pred}$ and observed $b_{abs,BrC}$ was 0.66. The symbol "*" indicates that the estimated value was statistically not higher than zero at the significance level of 5%.*

**9. Figure 12: Please indicate the binwidth used for the boxes.**

The bin boundaries used for the boxes have been added to the figure caption:

Figure 12 caption:
*... and horizontal lines within the boxes indicate medians. For panel a, each bin width is 0.1, from 0.5 to 1.0, and for panel b, each bin width is 0.2, from 0 to 1.0. In complement...*

**10. In order to somewhat reduce the quite high number of Figures, the authors could consider to move Figure 3 to the SI, as it does not present any results, and to remove Figure 15, which just seems to be a visual repetition of Table 4.**

We really appreciate these thoughtful suggestions. After careful internal discussion, we decided to keep the figures as they are, following our belief that they add higher value to the paper as main figures by providing impactful visualizations of some important observations and results.

**11. SI: As the SI will not get further typesetting I recommend giving captions together with the Figures, instead of listing them separately.**

We agree with the reviewer that this change will make the Supplementary Material more easily readable. Therefore, we have adjusted the location of figures in the supplementary text.
* * *
**Interactive comment by Dr. Marco Pandolfi:**

**I would like to offer a few comments based on a quick reading.**

**In Section 3.2.2 the authors estimated the mass absorption efficiency (MAE) of different PMF ACSM factors using multivariate linear regression (MLR). Using PMF factors in the MLR, rather than individual chemical species, has the great advantage of providing the MAE of atmospheric particles taking into consideration their mixing state in the atmosphere. However, as reported by the authors, the number of papers presenting MAE (or mass scattering efficiency; MSE) of pollutant sources is rather scarce. Here, I would like to suggest the authors to cite another recent paper (Ealo et al., ACP, 2018) where both the MAE and MSE of different PMF sources were reported. The chemical speciated data used in Ealo et al. (2018) were obtained from chemical analysis of 24h filters. In Ealo et al. (2018) the highest MAE was calculated for the Traffic source (around 1.7 m2/g at 637 nm).**
**Moreover, Ealo et al. (2018) also reported the correlation between measured and modelled aerosol particle scattering (R2 = 0.85) and absorption (R2 = 0.76).**

**Ealo, M., Alastuey, A., Pérez, N., Ripoll, A., Querol, X., and Pandolfi, M.: Impact of aerosol particle sources on optical properties in urban, regional and remote areas**
**in the north-western Mediterranean, Atmos. Chem. Phys., 18, 1149-1169, https://doi.org/10.5194/acp-18-1149-2018, 2018.**

We thank Dr. Pandolfi for providing this thoughtful comment to improve our manuscript. The reference suggested has been added to the text in the following instances:

Line 629:

Other studies have also made use of multivariate linear regression to retrieve mass absorption efficiencies (Hand and Malm, 2007;Washenfelder et al., 2015; Ealo et al., 2018).

Line 650:

*... light absorption. As a point of comparison, Ealo et al. (2018) conducted a study in the north-western Mediterranean and found the highest mass absorption efficiencies, ranging from 0.9 to 1.7 $m^2$ $g^{-1}$ at 637 nm, for traffic and industrial sources. As another point of comparison...*

**References**

Andreae, M. O. and Gelencsér, A.: Black carbon or brown carbon? The nature of light-absorbing carbonaceous aerosols, Atmos. Chem. Phys., 6, 10, 3131-3148, https://doi.org/10.5194/acp-6-3131-2006, 2006.

Chakraborty, S., Schiro, K. A., Fu, R., and Neelin, J. D.: On the role of aerosols, humidity, and vertical wind shear in the transition of shallow-to-deep convection at the Green Ocean Amazon 2014/5 site, Atmos. Chem. Phys., 18, 15, 11135-11148, https://doi.org/10.5194/acp-18-11135-2018, 2018.

de Sá, S. S., Palm, B. B., Campuzano-Jost, P., Day, D. A., Hu, W., Isaacman-VanWertz, G., Yee, L. D., Brito, J., Carbone, S., Ribeiro, I. O., Cirino, G. G., Liu, Y. J., Thalman, R., Sedlacek, A., Funk, A., Schumacher, C., Shilling, J. E., Schneider, J., Artaxo, P., Goldstein, A. H., Souza, R. A. F., Wang, J., McKinney, K. A., Barbosa, H., Alexander, M. L., Jimenez, J. L., and Martin, S. T.: Urban influence on the concentration and composition of submicron particulate matter in central Amazonia, Atmos. Chem. Phys. Discuss., 2018, 1-56, https://doi.org/10.5194/acp-2018-172, 2018.

Draxler, R. and Hess, G.: An overview of the HYSPLIT_4 modeling system for trajectories, dispersion, and deposition, Aust. Met. Mag., 47, 295-308, https://doi.org/10.5194/acp-13-8607-2013, 1998.

Ealo, M., Alastuey, A., Pérez, N., Ripoll, A., Querol, X., and Pandolfi, M.: Impact of aerosol particle sources on optical properties in urban, regional and remote areas in the north-western Mediterranean, Atmos. Chem. Phys., 18, 2, 1149-1169, https://10.5194/acp-18-1149-2018, 2018.

Machado, L. A. T., Dias, M. A. F. S., Morales, C., Fisch, G., Vila, D., Albrecht, R., Goodman, S. J., Calheiros, A. J. P., Biscaro, T., Kummerow, C., Cohen, J., Fitzjarrald, D., Nascimento, E. L., Sakamoto, M. S., Cunningham, C., Chaboureau, J.-P., Petersen, W. A., Adams, D. K., Baldini, L., Angelis, C. F., Sapucci, L. F., Salio, P., Barbosa, H. M. J., Landulfo, E., Souza, R. A. F., Blakeslee, R. J., Bailey, J., Freitas, S., Lima, W. F. A., and Tokay, A.: The Chuva Project: how does convection vary across Brazil?, Bull. Am. Meteorol. Soc., 95, 9, 1365-1380, https://doi.org/10.1175/bams-d-13-00084.1, 2014.

Moosmüller, H., Chakrabarty, R. K., and Arnott, W. P.: Aerosol light absorption and its measurement: A review, J Quant Spectrosc Radiat Transf, 110, 11, 844-878, https://doi.org/10.1016/j.jqsrt.2009.02.035, 2009.

Nunes, A. M. P., Silva Dias, M. A. F., Anselmo, E. M., and Morales, C. A.: Severe Convection Features in the Amazon Basin: A TRMM-Based 15-Year Evaluation, Front. Earth Sci., 4, 37, https://doi.org/10.3389/feart.2016.00037, 2016.

Pauliquevis, T., Lara, L. L., Antunes, M. L., and Artaxo, P.: Aerosol and precipitation chemistry measurements in a remote site in Central Amazonia: the role of biogenic contribution, Atmos. Chem. Phys., 12, 11, 4987-5015, https://10.5194/acp-12-4987-2012, 2012.

Saturno, J., Ditas, F., Penning de Vries, M., Holanda, B. A., Pöhlker, M. L., Carbone, S., Walter, D., Bobrowski, N., Brito, J., Chi, X., Gutmann, A., Angelis, I. H. d., Machado, L. A. T., Moran-Zuloaga, D., Rüdiger, J., Schneider, J., Schulz, C., Wang, Q., Wendisch, M., Artaxo, P., Wagner, T., Pöschl, U., Andreae, M. O., and Pöhlker, C.: African volcanic emissions influencing atmospheric aerosols over the Amazon rain forest, Atmos. Chem. Phys., 18, 14, 10391-10405, https://doi.org/10.5194/acp-18-10391-2018, 2018.

Trebs, I., Lara, L. L., Zeri, L. M. M., Gatti, L. V., Artaxo, P., Dlugi, R., Slanina, J., Andreae, M. O., and Meixner, F. X.: Dry and wet deposition of inorganic nitrogen compounds to a tropical pasture site (Rondônia, Brazil), Atmos. Chem. Phys., 6, 2, 447-469, https://10.5194/acp-6-447-2006, 2006.

Wang, X., Heald, C. L., Sedlacek, A. J., de Sá, S. S., Martin, S. T., Alexander, M. L., Watson, T. B., Aiken, A. C., Springston, S. R., and Artaxo, P.: Deriving brown carbon from multiwavelength absorption measurements: method and application to AERONET and Aethalometer observations, Atmos. Chem. Phys., 16, 19, 12733-12752, https://doi.org/10.5194/acp-16-12733-2016, 2016.

Zotter, P., Herich, H., Gysel, M., El-Haddad, I., Zhang, Y., Močnik, G., Hüglin, C., Baltensperger, U., Szidat, S., and Prévôt, A. S. H.: Evaluation of the absorption Ångström exponents for traffic and wood burning in the Aethalometer-based source apportionment using radiocarbon measurements of ambient aerosol, Atmos. Chem. Phys., 17, 6, 4229-4249, https://10.5194/acp-17-4229-2017, 2017.

---

## Author Response (AR1)

**Response to reviews**

Reviewer comments are in **bold**. Author responses are in plain text. Excerpts from the manuscript are in *italics*. Modifications to the manuscript are in *blue italics*. Page and line numbers in the responses correspond to those in the ACPD paper.

**Review #1**

**1. Line 155. As reported by (Zotter et al 2017) at 370 nm there maybe a nonnegligible light-absorption contribution from SOA compounds, I would use the wavelength at 430 nm.**

In our manuscript, the assessment of brown carbon absorption aimed exactly at quantifying the contribution of all organic material, including SOA, to light absorption. Therefore, the choice of the lower bound of wavelength measured (370 nm) followed as the most appropriate.

Previous studies in Amazonia also used the 370 nm wavelength to estimate the brown carbon contribution to particle light absorption (e.g., Wang et al., 2016; Saturno et al., 2018). Zotter et al. (2017) had a different goal compared to our study, aiming to apportion the contribution of traffic and wood burning aerosols to light absorption, excluding the possible contribution of SOA to light absorption.

**2. Line 166-170. I would use some references to support the assumptions made.**

We appreciate this idea, and have added the following references in the text:

Line 166:

"... (1) babs, BC(700) 166 = babs(700) and babs, BC(880) = babs(880) assuming that babs, BrC = 0 at red wavelengths (e.g., Andreae and Gelencsér, 2006; Wang et al., 2016), (2) aabs, BC(700, 880) was calculated from Equation 2 using babs, BC(700) and babs, BC(880), (3) babs, BC(370) was calculated from Equation 2, using babs, BC(880) and aabs, BC(370, 880) = aabs, BC(700, 880) under the assumption that aabs, BC was independent of wavelength (e.g., Andreae and Gelencsér, 2006; Moosmüller et al., 2009), and finally (4) babs, BrC(370) was obtained by Equation 1 using babs, BC(370) and babs(370).

**3.** Line **528-537**. When you describe Figure 11 you don't comment on the Angström exponent**

We thank the reviewer for catching this fault in completeness. We have adjusted the text as follows:

Line 534:

"... from 0.2 in the day to 0.4 at night. The absorption Angström exponent  $a_{abs}$  followed a similar diel trend, on average ranging from 2 during the day to 3 during the night (Figure 11d). Compared to the diel trends....

**4. Figure 4 and Figure 11: I would write the values of the interquartile ranges:**

**25, 75 or 10, 90?**

Our understanding is that an interquartile range is, by definition, the middle 50%, i.e. from 25% to 75%. Therefore, we have opted to keep the captions short.

**5. Figure 11: I would write the wavelengths also on the graph to be clear, in particular on the Angstrom exponent**

We appreciate this suggestion. However, in the interest of keeping the figure clean, the wavelengths are not added to the graph. We believe that this clarity is provided in the figure caption, where all wavelengths are explicitly mentioned. Please also note that all the variables plotted were explicitly defined in the Section 2.2 (Brown carbon light absorption), lines 173-174.

**6. Figure 13: I would draw the correlation curve and write the correlation coefficient as in Figure 14.**

To address this great suggestion by the reviewer, we have modified the figure to include the trend lines. The Pearson R values and the equation coefficients are provided in the figure caption.

**Review #2**

De Sá et al. have studied concentrations and light absorption properties of PM during the dry season in central Amazonia, as part of the GoAmazon2014/15 campaign. They present a wealth of data and analyze it in a comprehensive and detailed fashion to derive some interesting insights on anthropogenic impacts on PM and OA concentrations as well as light absorption properties over Amazonia. The paper is well written, the Figures are numerous, but clear and mostly justified (see comment below) and the conclusions are well-based on the measured data. The abstract could be improved (see comment below) and I have a few more comments listed below. I recommend publication of the paper after these have been addressed.

We thank the reviewer for reading our manuscript and providing this valuable feedback.

**Comments:**

**1.** The abstract is rather descriptive and would benefit from more detailed quantitative results, e.g. on the measured concentrations, PMF factor

contributions to OA, and contribution to BrC. Quantitative results would also be important to better understand some of the core findings mentioned in the abstract, e.g. on the BrC bleaching (L13), the relevance of sources other than BB (L17-19), and the suggested different oxidation pathways in the different clusters (L29-31). In turn, the parts that just describe what has been done could be condensed (e.g. L5-10, L13-15, L22-26, ...)

This feedback is highly appreciated. We have adjusted the abstract as follows:

[revised manuscript text omitted]

**2. L142: Calculating a new trajectory every 12 min seems like quite a high frequency to me. Do they change at all within such short time? Also, here it says that 48 h back trajectories were calculated while the caption of Figure 9 says 10 h.**

We thank the reviewer for bringing up these points. We have modified the text to clarify them as follows:

**Line 141:**

Air-mass backtrajectories were estimated using HYSPLIT4 (Draxler and Hess, 1998). Data sets of the S-band radar of the System for Amazon Protection (SIPAM) in Manaus (Machado et al., 2014) provided precipitation data, which allowed to filter out trajectories that intercepted precipitation. The HYSPLIT4 sSimulations started at 100 m above T3 and were calculated up to two days back in time for every 12 min to match with the radar data-up to two days back in time. Input meteorological data to the simulations were obtained on a grid of  $0.5^{\circ} \times 0.5^{\circ}$  were obtained from the Global Data Assimilation System (GDAS). Precipitation along the trajectories was based ondata sets of the S-band radar of the System for Amazon Protection (SIPAM) in-Manaus (Machado et al., 2014). Additional information on the backtrajectory calculations and on the radar were described in de Sá et al. (2018).

**Figure 9 caption:**

Trajectories were calculated using HYSPLIT4 in steps of 12 min and are shown for 10 h (Draxler and Hess, 1998).

**3.** L171-174: I cannot follow here. Are all the parameter subscripts correct? If so, please be more specific.**

We really appreciate that the reviewer caught this typo on the subscripts. We have corrected the issue as follows:

Line 172: *Based on babs.BC*(370) and *babs.BC*(430), *åabs*(370,430) was estimated.

**4. L209: "highly correlated" is not very precise. Please provide Persons r. Also, if OA and sulfate are really "highly correlated", does it imply common sources?**

Pearson R values were added to the text as shown below. In addition, we did not mean to say that organic and sulfate are highly correlated to each other. Rather, we meant that each of those species have their concentrations well correlated across sites. The text was adjusted to eliminate this ambiguity:

Line 209:

The time series of organic and sulfate mass concentrations across the three sites were well-highly correlated across over the two months when considering the timescale of a day (Figure 2a; 0.55 < R < 0.85). Similar behavior was observed for sulfate mass concentrations (Figure 2b; 0.86 < R < 0.93). The T0 and T3...

**5. L211-212: I have difficulties resolving timescales of less than a day in Figure 2. Please also give r.**

The R values were added to the text as follows:

Line 211:

The figure data also shows that for timescales of an hour less than a day the sites were less correlated (0.70 < R < 0.80 for sulfate, and 0.38 < R < 0.75 for organic mass concentrations).

6. L409: It is only at the very end of the discussion on possible drivers of lower concentrations during the wet season, that wet deposition is cautiously mentioned. To me this seems to be the most obvious and maybe also most relevant factor, as it efficiently removes both particles and precursor gases. Are there studies quantifying the effect of wet deposition in the area?

The reviewer brings up an excellent point about the way we presented our arguments. We reorganized the text as follows below, bringing the sentences on wet deposition to the beginning of the paragraph, in order to reflect its importance in the discussion. We also added a few more references that show different deposition patterns during the wet and dry seasons.

While it is known that wet deposition is important in regulating atmospheric concentrations in Amazonia, we are not aware of any quantitative field or modelling study on the effect of wet deposition on said concentrations. The challenge arises from the confounding nature of all the processes and sources that change simultaneously between the wet and dry seasons, making it difficult to apportion the reasons for differences in aerosol concentrations.

**Line 395:**

Therefore, reasons other than increased biomass burning in the dry season must have played a role in increasing organic PM1 concentrations. Importantly, the mass concentrations of sulfate and ammonium also increased by six-fold between seasons (Figure S10), indicating that atmospheric physical processes governing particle mass concentrations possibly played an important role. In this context, reduced wet deposition due to reduced convection in the dry season may be another appreciable one important contributor to the organic PM1 increases (Machado et al., 2014; Nunes et al., 2016; Chakraborty et al., 2018; Trebs et al., 2006; Pauliquevis et al., 2012). Another One-aspect is that BVOC emissions are typically higher...

**7. L729-743: These paragraphs in the Summary seem to add new aspects to the discussion of BrC that were not addressed before. I think they would fit better into the Results section, which would also help to shorten the quite long Summary section.**

We really like this suggestion and have moved the referred paragraph (lines 729-743) to the end of Section 3.2.2:

Line 689:

... and is discussed in the Supplementary Material (Figure S15). The BrC light absorption can have direct and indirect effects on radiative forcing...

**8. Table 3: Statistical significance is mentioned in the caption, please include the significance level used (alpha = 0.05?) and ideally also the p-values of the model coefficients (i.e. Eabs).**

We have adjusted the caption to address this suggestion. Note that the confidence intervals provided for  $E_{abs}$  values were stated as 95%. It thus follows that the significance level is 5%, and we have now explicitly added that to the caption. We have also reiterated from the text that the values listed were found by running the constrained least squares regression in bootstrap:

Table 3. Results of the constrained linear least squares regression analysis for the brown-carbon absorption coefficient (Equation 5). (a) Mass absorption efficiency  $E_{abs}$  associated with each PMF factor. (b) Model intercept. The mean, standard error (SE), and 95% confidence interval (CI) are listed in each panel. They were obtained through bootstrap of the regression analysis considering different samples (i.e., sets of points in time) for 104 runs. Unit of Mm-1 represents 10-6 m-1. The coefficient of determination R2 between predicted  $b_{abs,BrC,pred}$  and observed  $b_{abs,BrC}$  was 0.66. The symbol "\*" indicates that the estimated value was statistically not higher than zero at the significance level of 5%.

**9. Figure 12: Please indicate the binwidth used for the boxes.**

The bin boundaries used for the boxes have been added to the figure caption:

**Figure 12 caption:**

... and horizontal lines within the boxes indicate medians. For panel a, each bin width is 0.1, from 0.5 to 1.0, and for panel b, each bin width is 0.2, from 0 to 1.0. In complement...

**10. In order to somewhat reduce the quite high number of Figures, the authors could consider to move Figure 3 to the SI, as it does not present any results, and to remove Figure 15, which just seems to be a visual repetition of Table 4.**

We really appreciate these thoughtful suggestions. After careful internal discussion, we decided to keep the figures as they are, following our belief that they add higher value to the paper as main figures by providing impactful visualizations of some important observations and results.

**11. SI: As the SI will not get further typesetting I recommend giving captions together with the Figures, instead of listing them separately.**

We agree with the reviewer that this change will make the Supplementary Material more easily readable. Therefore, we have adjusted the location of figures in the supplementary text.

**Interactive comment by Dr. Marco Pandolfi:**

I would like to offer a few comments based on a quick reading.

In Section 3.2.2 the authors estimated the mass absorption efficiency (MAE) of different PMF ACSM factors using multivariate linear regression (MLR). Using PMF factors in the MLR, rather than individual chemical species, has the great advantage of providing the MAE of atmospheric particles taking into consideration their mixing state in the atmosphere. However, as reported by the authors, the number of papers presenting MAE (or mass scattering efficiency; MSE) of pollutant sources is rather scarce. Here, I would like to suggest the authors to cite another recent paper (Ealo et al., ACP, 2018) where both the MAE and MSE of different PMF sources were reported. The chemical speciated data used in Ealo et al. (2018) were obtained from chemical analysis of 24h filters. In Ealo et al. (2018) the highest MAE was calculated for the Traffic source (around 1.7 m2/g at 637 nm).

Moreover, Ealo et al. (2018) also reported the correlation between measured and modelled aerosol particle scattering (R2 = 0.85) and absorption (R2 = 0.76).

Ealo, M., Alastuey, A., Pérez, N., Ripoll, A., Querol, X., and Pandolfi, M.: Impact of aerosol particle sources on optical properties in urban, regional and remote areas

in the north-western Mediterranean, Atmos. Chem. Phys., 18, 1149-1169, https://doi.org/10.5194/acp-18-1149-2018, 2018.

We thank Dr. Pandolfi for providing this thoughtful comment to improve our manuscript. The reference suggested has been added to the text in the following instances:

Line 629:

Other studies have also made use of multivariate linear regression to retrieve mass absorption efficiencies (Hand and Malm, 2007;Washenfelder et al., 2015; Ealo et al., 2018).

Line 650:

... light absorption. As a point of comparison, Ealo et al. (2018) conducted a study in the north-western Mediterranean and found the highest mass absorption efficiencies, ranging from 0.9 to  $1.7 \text{ m}^2 \text{ g}^{-1}$  at 637 nm, for traffic and industrial sources. As another point of comparison...

**1 Abstract**

2 Urbanization and deforestation have important impacts on atmospheric particulate matter 3 (PM) over Amazonia. This study presents observations and analysis of submicron PM1 4 concentration, composition, and optical properties in central Amazonia during the dry season, 5 focusing on the anthropogenic impacts. The focus is on delineating the anthropogenic impact on 6 the observed quantities, especially as related to the organic PM1. The primary study site was 7 located 70 km downwindto the west of Manaus, a city of over two million people in Brazil, as-8 As part of the GoAmazon2014/5 experiment., datasets from a large suite of instrumentation were 9 employed. A high-resolution time-of-flight aerosol mass spectrometer (AMS) provided data on 10 PM1 composition, and aethalometer measurements were used to derive the absorption coefficient 11 babs,BrC of brown carbon (BrC) at 370 nm. Non-refractory PM1 mass concentrations averaged 12 12.2  $\mu$ g m-3 at the primary study site, dominated by organics (83%) and sulfate (11%). A 13 decrease in  $b_{abs,BrC}$  was observed as the mass concentration of nitrogen-containing organic 14 compounds decreased and the organic PM1 O:C ratio increased, suggesting atmospheric bleaching of the BrC components. The relationships of babs. BrC with AMS-measured quantities 15 16 showed that the absorption was associated with less-oxidized, nitrogen-containing organic 17 compounds. Atmospheric processing appeared to bleach the BrC components. The organic PM1 18 was separated into six different classes by positive-matrix factorization (PMF), and the mass 19 absorption efficiency Eabs associated with each factor was estimated through multivariate linear 20 regression of  $b_{abs,BrC}$  on the factor loadings. Estimates of the effective mass absorption efficiency associated with each PMF factor were obtained. The largest Eabs values were associated with 21 urban  $(2.04 \pm 0.14 \text{ m}^2 \text{ g}^{-1})$  and biomass burning  $(0.82 \pm 0.04 \text{ m}^2 \text{ g}^{-1} \text{ to } 1.50 \pm 0.07 \text{ m}^2 \text{ g}^{-1})$ 22 23 sources. Together, these sources Biomass burning and urban emissions appeared to contributed 24 at least 80% of  $b_{\rm 
[revised manuscript text omitted]
_{\text{abs,BrC}} = b_{\text{abs,Org}_1} + b_{\text{abs,Org}_2} + \dots + b_{\text{abs,Org}_n}$$
(3)

650 The treatment assumes the absence of cross-interactions among the parts and holds for a single 651 wavelength. The absorption coefficient  $b_{abs,i}$  of part *i* is defined as follows:

 $b_{\text{abs},i} = E_{\text{abs},i} \times C_i \tag{4}$

where  $E_{abs,i}$  is the mass absorption efficiency and  $C_i$  is the mass concentration of part *i*. Based on equations 33 and 44, the following model was constructed for  $b_{abs,BrC}$  by using the PMF factor loadings as a proxy for the mass concentrations of organic PM1 components:

656
$$b_{abs,BrC} = E_{abs,MO-OOA} G_{MO-OOA} + E_{abs,LO-OOA} G_{LO-OOA} + E_{abs,IEPOX-SOA} G_{IEPOX-SOA} + E_{abs,MO-BBOA} G_{MO-BBOA} + E_{abs,LO-BBOA} G_{LO-BBOA} + E_{abs,HOA} G_{HOA} + B$$
(5)

[revised manuscript text omitted]
     | $0.24\pm0.01$ | < 0.001         | $1.20\pm0.10$ | $1.25\pm0.08$ | 1.00                                             |
| LO-OOA     | $0.15\pm0.01$ | $0.001\pm0.001$ | $0.86\pm0.08$ | $1.51\pm0.06$ | 0.99                                             |
| IEPOX-SOA  | $0.14\pm0.01$ | < 0.001         | $0.74\pm0.02$ | $1.51\pm0.01$ | 0.99                                             |
| MO-BBOA    | $0.13\pm0.01$ | $0.011\pm0.003$ | $0.70\pm0.07$ | $1.59\pm0.11$ | N/A                                              |
| LO-BBOA    | $0.02\pm0.01$ | $0.05\pm0.01$   | $0.53\pm0.04$ | $1.79\pm0.06$ | N/A                                              |
| НОА        | $0.05\pm0.01$ | $0.001\pm0.001$ | $0.22\pm0.03$ | $1.82\pm0.03$ | 0.94                                             |

**Table 2.** Relationship of PMF factors to organo-nitrogen characteristics. Listed for each factor are the mean loading of the time series, the percent contribution of the  $C_xH_yO_zN_p^+$  family to the factor profile, the mean mass concentration of the  $C_xH_yO_zN_p^+$  family (based on multiplication of columns 2 and 3), as well as the Pearson-*R* correlation of factor loading against the mass concentration of  $C_xH_yO_zN_p^+$ , the mass concentration of organic nitrates, and  $b_{abs,BrC}$ . The  $C_xH_yO_zN_p^+$  family corresponds to the sum of all ions containing at least one C atom and one N atom, as measured by the AMS. Detailed family-colored spectra showing the nitrogencontaining ions for all PMF factors are presented in Figure S6, and the most important ion fits are shown in Figure S7. The AMS method characterizes organic nitrates through the NO+ and NO+ fragments, which remain distinct from the larger fragments of the  $C_xH_yO_zN_p^+$  family (Section S1 and discussion therein).

|               |                                                       | Nitrogen characteristics of factor profile        |                                                                                        | Pearson R of factor loading                                       |                                                                |                                 |
|---------------|-------------------------------------------------------|---------------------------------------------------|----------------------------------------------------------------------------------------|--------------------------------------------------------------------------|----------------------------------------------------------------|---------------------------------|
| PMF
factor | Mean
factor
loading
(µg
m -3 ) | $C_xH_yO_zN_p^+$
family
contribution
(%) | Mass
concentration
of the
$C_xH_yO_zN_p^+$
family (µg
m -3 ) | Against the
mass
concentration
of
$C_xH_yO_zN_p^+$
family | Against the
mass
concentration
of organic
nitrates | Against
b abs,BrC |
| MO-
OOA    | 1.6                                                   | 5.7                                               | 0.09                                                                                   | 0.33                                                                     | 0.38                                                           | 0.17                            |
| LO-
OOA    | 2.2                                                   | 3.7                                               | 0.08                                                                                   | 0.10                                                                     | 0.15                                                           | -0.19                           |
| IEPOX-
SOA | 1.2                                                   | 6.6                                               | 0.08                                                                                   | 0.39                                                                     | 0.40                                                           | 0.17                            |
| MO-
BBOA   | 1.5                                                   | 2.9                                               | 0.04                                                                                   | 0.65                                                                     | 0.24                                                           | 0.53                            |

| LO-
BBOA | 1.0 | 10.4 | 0.11 | 0.89 | 0.13 | 0.69 |
|-------------|-----|------|------|------|------|------|
| HOA         | 0.6 | 9.0  | 0.05 | 0.82 | 0.20 | 0.68 |

**Table 3.**Results of the constrained linear least squares regression analysis for the brown-
carbon absorption coefficient (Equation 55). (a) Mass absorption efficiency  $E_{abs}$
associated with each PMF factor. (b) Model intercept. The mean, standard error
(SE), and 95% confidence interval (CI) are listed in each panel. They were obtained
through bootstrap of the regression analysis considering different samples (i.e., sets
of points in time) for 104 runs. Unit of Mm-1 represents 10-6 m-1. The coefficient of
determination  $R^2$  between predicted  $b_{abs,BrC,pred}$  and observed  $b_{abs,BrC}$  was 0.66. The
symbol "\*" indicates that the estimated value was statistically not higher than zero at
the significance level of 5%.

| (a)             | $E_{\rm abs}({ m m}^2~{ m g}^{-1})$ |      |              |  |
|-----------------|-------------------------------------|------|--------------|--|
| PMF factors     | Mean                                | SE   | CI           |  |
| MO-OOA          | 0.01*                               | 0.02 | [0.00, 0.08] |  |
| LO-OOA          | 0.01*                               | 0.02 | [0.00, 0.08] |  |
| IEPOX-SOA       | 0.40                                | 0.05 | [0.31, 0.50] |  |
| MO-BBOA         | 0.82                                | 0.04 | [0.75, 0.90] |  |
| LO-BBOA         | 1.50                                | 0.07 | [1.37, 1.63] |  |
| НОА             | 2.04                                | 0.14 | [1.76, 2.31] |  |
|                 |                                     |      |              |  |
| (b)             | $b_{\rm abs}({\rm Mm}^{-1})$        |      |              |  |
| Model intercept | Mean                                | SE   | CI           |  |

| B 0.13* 0. | 10 [0.00, 0.33] |
|-------------------|-----------------|
|-------------------|-----------------|

Table 4. Contribution of PM1 components as represented by the PMF factors to organic mass concentrations and BrC light absorption. The contribution of the model intercept to BrC light absorption is also included. Values listed are resulting means and standard deviations of the contributions calculated throughout IOP2. Small differences between the values in column 2 and the values represented in the inset of Figure 55a are due to differences in data coverage by the aethalometer and AMS.

| PMF factor      | Contribution to organic | Contribution to BrC  |
|-----------------|-------------------------|----------------------|
|                 | mass concentration (%)  | light absorption (%) |
| MO-OOA          | $21.1 \pm 10.0$         | $0.5\pm0.4$          |
| LO-OOA          | $30.9 \pm 11.4$         | $0.8\pm0.5$          |
| IEPOX-SOA       | $16.3\pm9.8$            | $15.7 \pm 11.2$      |
| MO-BBOA         | $16.7\pm12.0$           | $28.9\pm18.0$        |
| LO-BBOA         | $9.5\pm7.5$             | $27.8 \pm 14.3$      |
| HOA             | $5.5\pm3.9$             | $21.7\pm10.5$        |
| Model intercept | N/A                     | $4.6\pm2.6$          |

**List of Figures**

[revised manuscript text omitted]